# NMDA receptors in visual cortex are necessary for normal visuomotor integration and skill learning

**Felix C Widmer[1,2], Sean M O'Toole[1], Georg B Keller[1,2]\***

[1]Friedrich Miescher Institute for Biomedical Research, Basel, Switzerland; [2]Faculty of Science, University of Basel, Basel, Switzerland

**Abstract** The experience of coupling between motor output and visual feedback is necessary for the development of visuomotor skills and shapes visuomotor integration in visual cortex. Whether these experience-dependent changes of responses in V1 depend on modifications of the local circuit or are the consequence of circuit changes outside of V1 remains unclear. Here, we probed the role of $N$-methyl-D-aspartate (NMDA) receptor-dependent signaling, which is known to be involved in neuronal plasticity, in mouse primary visual cortex (V1) during visuomotor development. We used a local knockout of NMDA receptors and a photoactivatable inhibition of CaMKII in V1 during the first visual experience to probe for changes in neuronal activity in V1 as well as the influence on performance in a visuomotor task. We found that a knockout of NMDA receptors before, but not after, first visuomotor experience reduced responses to unpredictable stimuli, diminished the suppression of predictable feedback in V1, and impaired visuomotor skill learning later in life. Our results demonstrate that NMDA receptor-dependent signaling in V1 is critical during the first visuomotor experience for shaping visuomotor integration and enabling visuomotor skill learning.

## Editor's evaluation

This study demonstrates that the development of visuomotor mismatch signals in V1 as well as behavioral reactions to mismatches depend on NMDA receptors in V1 during early visual experience, suggesting a critical role of NMDA receptor-dependent plasticity within V1 in forming internal models that transform self-movement to visual experiences. The result will be of great interest to the neuroscience community.

**\*For correspondence:** georg.keller@fmi.ch

**Competing interest:** The authors declare that no competing interests exist.

## Introduction

Movement results in predictable sensory consequences. Through experience, the brain learns this transformation from motor output to sensory feedback. These transformations between different coding coordinate systems are referred to as internal models and are essential for the capacity of using sensory input to guide movements (*Jordan and Rumelhart, 1992*). When raised without coupling between movements and sensory feedback during visual development, kittens fail to use visual input to guide movements (*Held and Hein, 1963*). The extent of the impairment is specific to the movement that is experimentally uncoupled from visual feedback. When kittens are reared with a neck collar that prevents them from seeing the effect of moving their paws independently, but allows them to see the effect of extending their paws while standing, they have normal visual approach paw-extension reflexes, but fail to perform visually guided reaches (*Hein and Held, 1967*). Coupling between locomotion and visual feedback is also necessary to integrate visual and motor-related signals in primary visual cortex (V1). Under normal conditions, distinct and salient responses have been observed in V1

following unpredictable mismatches between movement and visual feedback in both humans and mice (*Keller et al., 2012*; *Stanley and Miall, 2007*; *Zmarz and Keller, 2016*). In mice raised from birth without coupling between movement and visual feedback, visuomotor mismatch responses are absent and only emerge after first exposure to normal visuomotor coupling (*Attinger et al., 2017*). Thus, the coupling between movement and visual feedback is essential for both visuomotor behavior and normal visuomotor integration of neuronal activity in V1.

Given that V1 receives both the bottom-up visual input and signals consistent with a top-down prediction of visual feedback given movement (*Leinweber et al., 2017*) necessary to compute mismatch responses, it has been speculated that mismatch responses are computed locally in V1. Neurons in layer 2/3 (L2/3) of V1 that are responsive to visuomotor mismatch receive balanced and opposing top-down motor-related and bottom-up visual input (*Jordan and Keller, 2020*). This is consistent with a subtractive computation of mismatch responses, and it is conceivable that L2/3 more generally functions as a comparator between bottom-up and top-down signals. It has been postulated that visuomotor experience establishes this balance between top-down and bottom-up input on individual L2/3 neurons in V1 (*Hertäg and Sprekeler, 2020*). If this were so, we would predict that perturbing any of the essential subcellular mechanisms related to plasticity in V1 during visuomotor development would result in a reduction of mismatch responses.

Here, we tested this by interfering with plasticity in V1 during the first visuomotor experience as broadly as possible using two separate approaches. First, we used a local knockout of *N*-methyl-D-aspartate (NMDA) receptors in V1 prior to first visuomotor experience. NMDA receptors are involved in a wide variety of different forms of plasticity (*Paoletti et al., 2013*; *Rodriguez et al., 2019*) and are essential for activity-dependent synaptic strengthening in cortex (*Hasan et al., 2013*; *Kirkwood and Bear, 1994*; *Lo et al., 2013*). In a parallel approach, to impair NMDA receptor-dependent signaling in V1 in a cell-type-specific manner, we used a photoactivatable inhibitor of the calcium/calmodulin-dependent protein kinase II (CaMKII). CaMKII has been shown to be an essential element of NMDA receptor-dependent signaling (*Barria and Malinow, 2005*; *Gambrill and Barria, 2011*; *Wang et al., 2011*). NMDA receptors are thought to exert their influence on synaptic plasticity by increasing calcium influx into the cell, where calmodulin binds calcium and activates CaMKII. The idea that NMDA receptors and CaMKII are on the same plasticity pathway is supported by several findings. For example, spine enlargement triggered by NMDA receptor stimulation can be inhibited by blocking CaMKII (*Herring and Nicoll, 2016*). Additionally, activated CaMKII and NMDA receptors directly interact (*Leonard et al., 1999*) to form CaMKII-NMDA receptor complexes that are required for the induction of long-term potentiation (*Barria and Malinow, 2005*) and likely control synaptic strength (*Lisman et al., 2012*). Thus, we predicted that an NMDA receptor knockout and CaMKII inhibition would have similar effects on experience-dependent functional changes in L2/3 neurons.

## Results

### Knockout of *Grin1* prior to first visual experience impaired the development of normal visual and visuomotor mismatch responses

We first quantified the effect of a conditional knockout of NMDA receptors in V1 prior to first visual experience on the responses of L2/3 V1 neurons. To achieve this, we used *Grin1^tm2Stl* mice, which carry a modified version of the *Grin1* gene (coding for an essential subunit of the NMDA receptor) that can be rendered inactive by Cre recombination (*Tsien et al., 1996*). We dark-reared these mice from birth and injected an adeno-associated viral vector (AAV2/1-EF1α-Cre-T2A-mCherry) unilaterally into V1 to express Cre recombinase at postnatal day (P)21 prior to first visual experience (ΔGrin1_juv; *Figure 1A and B*). At P30, we injected a second AAV vector (AAV2/1-EF1α-GCaMP6f) to express GCaMP6f bilaterally in both primary visual cortices to record neuronal activity in the knockout hemisphere and a within-mouse control hemisphere. Mice were then exposed to visual input for the first time in their life at P32 when they were introduced to a virtual environment that provided closed-loop feedback between forward locomotion and backward visual flow in a virtual corridor (*Attinger et al., 2017*). Mice were trained in this setup for 2 hr every other day for 12 days (for a total of six sessions), after which we measured calcium activity in L2/3 neurons using two-photon imaging (*Figure 1C*). During the imaging experiments, mice were first exposed to closed-loop visual flow feedback in a virtual corridor (see Materials and methods). To measure mismatch responses, we introduced brief (1 s) halts

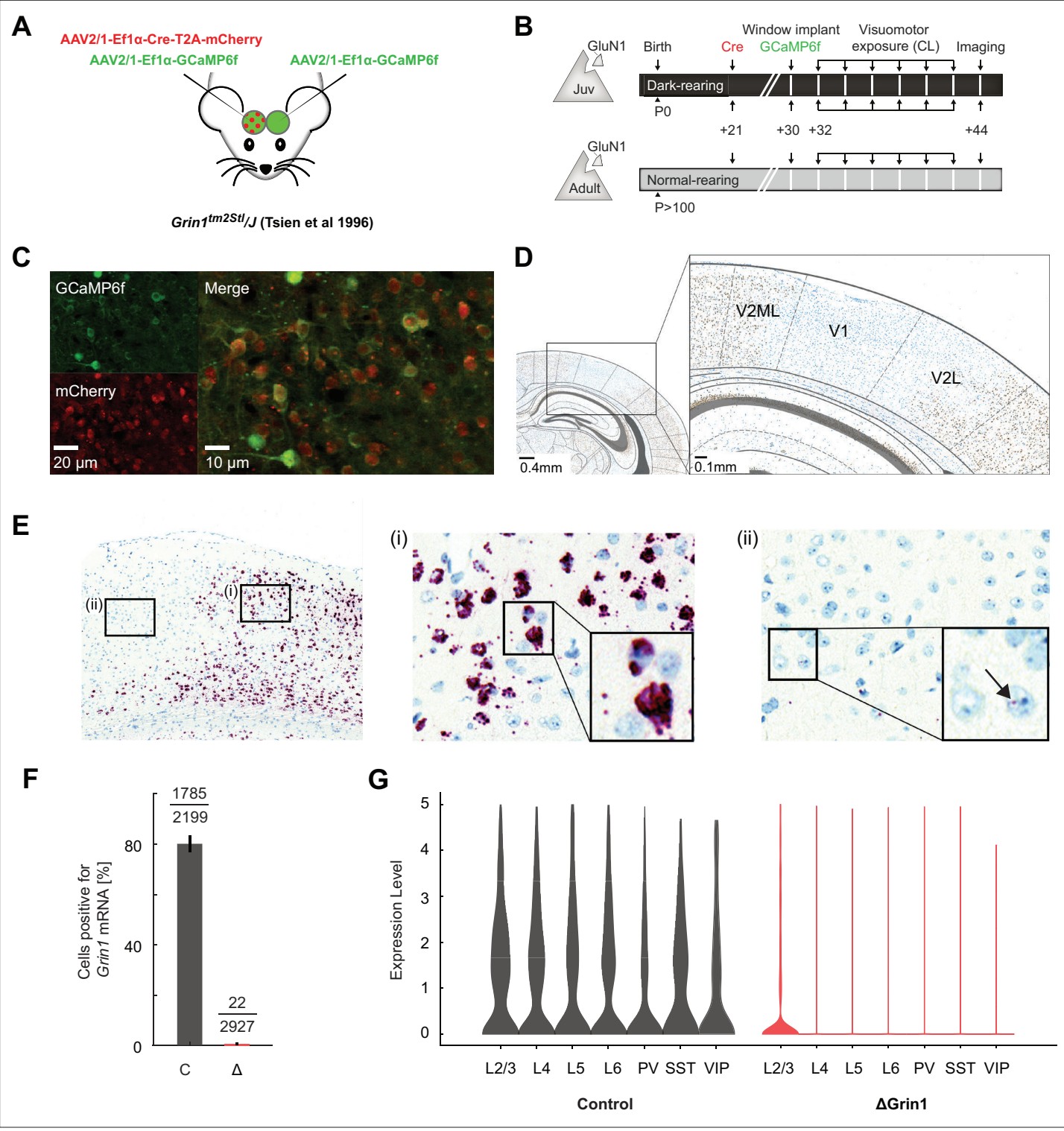

**Figure 1.** Characterization of *N*-methyl-ᴅ-aspartate (NMDA) receptor knockout. (**A**) We injected an adeno-associated viral vector (AAV) to express Cre recombinase unilaterally and another to express a calcium indicator bilaterally (GCaMP6f) in V1 of ΔGrin1 mice. (**B**) Experimental timeline: a first group of *Grin1* mice (ΔGrin1$_{juv}$) was dark-reared from birth. We injected an AAV to express Cre at postnatal day (P)21 unilaterally in V1, injected a second AAV bilaterally to express GCaMP6f, and implanted imaging windows bilaterally at P30. A second group of *Grin1* mice (ΔGrin1$_{adult}$) was reared normally and received the same injections at p>100. All mice then had six sessions of visuomotor exposure in a closed-loop (CL) virtual environment before imaging experiments. (**C**) Example two-photon images showing co-expression of GCaMP6f and Cre-mCherry constructs. (**D**) In situ hybridization against *Grin1* mRNA (see Materials and methods) confirming the local knockout of *Grin1* in V1. Blue: hematoxylin stain for cell nuclei; brown: *Grin1* hybridization

*Figure 1 continued on next page*

*Figure 1 continued*

signal. Brain regions were identified using a mouse brain atlas (***Franklin and Paxinos, 2012***). (**E**) Injection sites of Cre were readily visible in *Grin1* in situ hybridization images. Outside of the injection site (i), labeling was dense in most cells with multiple puncta per cell. In injection sites (ii), labeling was almost completely absent. Inset shows a cell with one punctum (arrow); if a cell had more than two of these puncta, it was counted positive in the analysis shown in (**F**). (**F**) The fraction of cells positive (more than two puncta per cell) for *Grin1* mRNA in 0.5 mm × 0.5 mm regions in injection sites (Δ) was strongly reduced compared to regions outside of injection sites (C). (**G**) *Grin1* knockout reduced expression of *Grin1* in all major cortical neuron types. Left: violin plots of the number of single-cell sequencing mRNA reads corresponding to the portion of the *Grin1* gene knocked out in the ΔGrin1 mice and control mice. Layer 2/3 (L2/3), layer 4 (L4), layer 5 (L5), and layer 6 (L6) excitatory neurons, and parvalbumin (PV)-positive, somatostatin (SST)-positive, and vasoactive peptide (VIP)-positive interneurons. Expression levels are normalized to the total number of reads per nuclei. Right: the same data for ΔGrin1 mice. Expression levels were significantly downregulated in all neuron types, with the exception of VIP neurons (see ***Supplementary file 1A*** for statistics).

The online version of this article includes the following figure supplement(s) for figure 1:

**Figure supplement 1.** Single-nuclei RNA-sequencing data covers all major cortical neuron types.

of visual flow at random times (***Keller et al., 2012***). To estimate the contributions of visual flow and locomotion separately, mice were then presented with a playback of the visual flow they had previously self-generated in the closed-loop condition while they were free to run on the spherical treadmill (we will refer to this as the open-loop condition). To measure visual responses, mice were presented with full-field drifting gratings of different orientations. Finally, to isolate motor-related signals, we measured locomotion-related activity in complete darkness.

We validated the method for the local knockout of *Grin1* using an in situ hybridization with a *Grin1* mRNA probe in a subset of mice and found a marked reduction in *Grin1* expression at the injection site of the Cre vector (***Figure 1D***). To quantify the knockout-induced reduction in *Grin1* expression and to test whether the knockout affects all neuron types, we first used the in situ hybridization images to quantify the fraction of *Grin1*-positive cells in randomly selected 500 μm × 500 μm regions at the Cre injection site and repeated the same quantification in the control hemisphere. Assuming the knockout occurs in only a subset of cells that normally express *Grin1*, we should find a fraction of cells that remain positive for *Grin1* in the knockout regions. However, the fraction of cells that exhibited even minimal evidence of *Grin1* expression was below 1% in the knockout region (***Figure 1E and F***), and thus the knockout is likely present in all cell types. To confirm this, we then used single-nuclei RNA sequencing to quantify *Grin1* expression levels. ΔGrin1 mice were injected at P21 with either Cre (knockout) or Flp (control) AAV vectors, both also driving expression of an mCherry fluorophore. Mice were sacrificed between P42 and P45, and V1 tissue was extracted and processed for single-nuclei RNA sequencing on the 10X Genomics platform (see Materials and methods). In both conditions, our sequencing data covered all of the major neuronal groups (***Figure 1—figure supplement 1***). *Grin1* expression levels were reduced in all neuron types, and except for VIP-positive interneurons, which was the group with the fewest neurons, this reduction was statistically significant (***Figure 1G***). Thus, the NMDA receptor knockout affects all neuron types in V1.

To determine the functional effect of the NMDA receptor knockout, we started by quantifying visuomotor mismatch responses in the closed-loop condition and found that in the knockout hemisphere of ΔGrin1$_{juv}$ mice mismatch responses were reduced compared to the control hemisphere (***Figure 2A***). This reduction was commensurate with the response reduction in mice that never experienced coupling between locomotion and visual flow (***Figure 2—figure supplement 1A***). We also found a reduction in the amplitude of grating onset responses (***Figure 2B***), but no evidence of a reduction in motor-related activity upon running onset in a closed-loop environment (***Figure 2C***). The fact that mismatch and visual responses are influenced by the NMDA receptor knockout is consistent with an impairment of the comparator function of L2/3 (***Jordan and Keller, 2020***). An alternative explanation would be that the reduced responses are simply a consequence of an overall reduction in activity levels in L2/3. However, this was not the case as comparing mean activity levels between control and knockout hemispheres revealed no evidence of a reduction in activity (***Figure 2D***).

Mismatch responses are thought to arise from a transient imbalance between opposing bottom-up visual inhibition and top-down motor-related excitation. A reduction of mismatch response could be the result of a reduction in either top-down or bottom-up input, or a failure to appropriately match bottom-up inhibition and top-down excitation. To disambiguate these possibilities, we estimated the contribution of bottom-up visual input and top-down motor-related

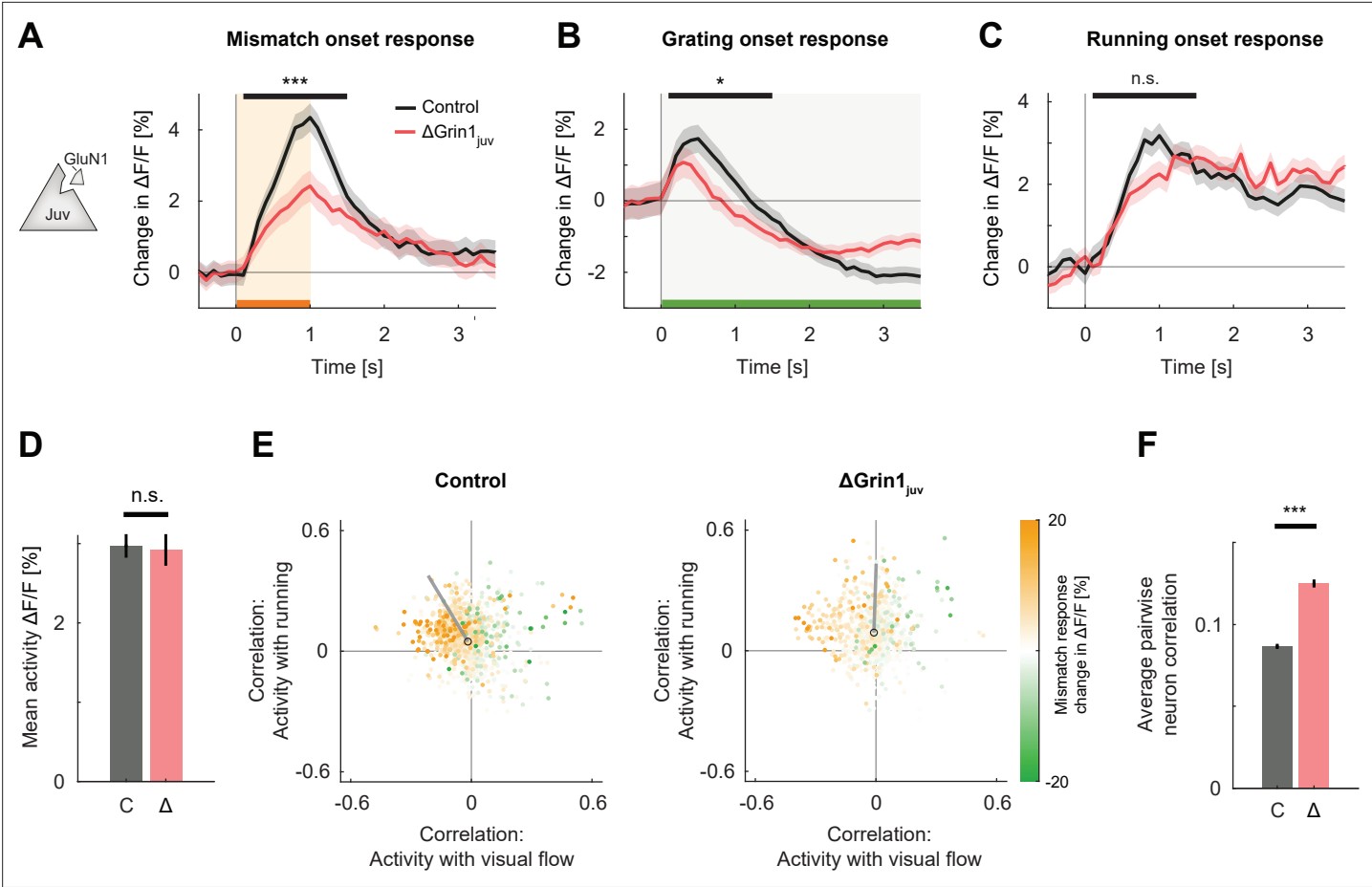

**Figure 2.** *N*-methyl-ᴅ-aspartate (NMDA) receptor knockout prior to first visual experience impaired the development of normal visual and visuomotor mismatch responses. (**A**) The average L2/3 population response to mismatch was stronger in control (black) than in ΔGrin1$_{juv}$ (red) hemispheres. Shading indicates the standard error of the mean (SEM) across neurons. Orange shading and bar indicate the duration of mismatch. Mean responses were compared across neurons in the time window indicated by the black bar above the traces. Here and in subsequent panels, n.s.: p>0.05, *p<0.05, **p<0.01, ***p<0.001. For all details of statistical testing, see *Supplementary file 1A*. (**B**) As in (**A**), but for responses to the onset of a drifting grating stimulus (see Materials and methods). Green shading and bar indicate the presence of a grating stimulus. (**C**) As in (**A**), but for running onset responses in the closed-loop condition. (**D**) Mean calcium activity of neurons in the control (**C**, gray) and ΔGrin1$_{juv}$ (Δ, red) hemisphere during the closed-loop condition. Error bars indicate SEM across neurons. (**E**) Scatter plot of the correlation between neuronal activity and visual flow, and the correlation between neuronal activity and running speed in the open-loop condition for all L2/3 neurons recorded in control (left) and ΔGrin1$_{juv}$ (right) hemispheres. Each dot shows the correlations for one neuron, and dot color indicates the neuron's mismatch response. Black circles mark the population mean, and solid gray lines indicate the direction of the first principal component of the distribution (see *Figure 2—figure supplement 1B* and Materials and methods). (**F**) Average pairwise correlation of neuronal activity was higher in ΔGrin1$_{juv}$ (Δ, red) compared to that in the control (**C**, gray) hemisphere. Error bars indicate SEM across neurons.

The online version of this article includes the following figure supplement(s) for figure 2:

**Figure supplement 1.** The effect of the *N*-methyl-ᴅ-aspartate (NMDA) receptor knockout was comparable to the lack of experience with visuomotor coupling or systemic block of NMDA receptors.

input separately. We did this in the open-loop condition by calculating the correlation between neuronal activity and visual flow, and that between neuronal activity and locomotion for each neuron (*Figure 2E*). The correlation pattern in the control hemisphere was consistent with previously published data (*Attinger et al., 2017*), in that neurons with large mismatch responses tended to show a negative correlation with visual flow and a positive correlation with running speed. In the knockout hemisphere, we found that the overall distribution was comparable to that observed in mice raised without coupling between running and visual flow (*Attinger et al., 2017*). We quantified this using the angle of the first principal component of the distribution relative to the axis defined by the correlation of activity with running speed. This metric quantifies the overall relative

influence of running speed and visual flow on the population of L2/3 neurons and has been shown to be sensitive to whether a mouse had experience with coupling between locomotion and visual flow during development (*Attinger et al., 2017*). A principal component of this distribution close to the positive diagonal would be consistent with an additive integration of running speed and visual flow signals, while a principal component along the negative diagonal would be consistent with a subtractive integration of running speed and visual flow signals. Similar to the distribution observed in mice raised with coupling between running and visual flow, we found that in the control hemisphere the majority of neurons exhibited opposing signs of correlation with running and visual flow, which manifested as a principal component close to the negative diagonal. In the knockout hemisphere, the distribution was shifted in the direction of that observed in mice raised without coupling between running and visual flow, where the principal component is rotated towards the positive diagonal (*Figure 2—figure supplement 1B*). These results are consistent with the interpretation that the NMDA receptor knockout interferes with the establishment of the balance between opposing top-down and bottom-up input in individual neurons. Lastly, consistent with the effect of systemic inhibition of NMDA receptors on correlations of activity between L2/3 neurons in V1 (*Figure 2—figure supplement 1D*; *Hamm et al., 2017*), we found that in the knockout hemisphere the average pairwise correlation of neuronal activity was higher compared to that in the control hemisphere (*Figure 2F*). Thus, the NMDA receptor knockout prior to first visual experience had effects on local activity commensurate with NMDA receptor inhibition and prevented the development of normal visual and visuomotor mismatch responses in V1.

These results would be consistent with either a role of the NMDA receptor in the plasticity necessary for the establishment of visuomotor mismatch responses in V1 or a direct involvement of NMDA receptors in generating neuronal calcium responses. The latter could be driven by an influence of NMDA receptors on the overall excitability of the neurons or, given that NMDA receptors conduct calcium, by directly reducing the calcium response. To disambiguate this, we repeated the same NMDA receptor knockout experiments in a second group of mice that had been reared in a normal light-dark cycle and received a *Grin1* knockout as adults (>100 days), 9 days prior to the start of the training sessions ($\Delta Grin1_{adult}$; *Figure 1B*). We found that in these mice there was no difference in mismatch, grating, or running onset responses between those in the control hemisphere and those in the knockout hemisphere (*Figure 3A–C*). However, consistent with the finding that pharmacological inhibition of NMDA receptors in adult mice results in an overall decrease of V1 activity (*Ranson et al., 2019*; *Figure 2—figure supplement 1C*), we found a reduction in overall activity levels in the knockout hemisphere (*Figure 3D*). Consistent with a lack of an NMDA receptor knockout-induced change in mismatch and visual responses, the distribution of visual flow and running correlations with activity in control and knockout hemispheres was similar (*Figure 3E*). Lastly, as in the juvenile knockout, we found an increase in the average correlation between neurons (*Figure 3F*). This increase in correlation is likely specific to L2/3 neurons, as a similar knockout in layer 4 (L4) neurons results in a decrease in correlation between neurons that lack NMDA receptors (*Mizuno et al., 2021*). This demonstrates that NMDA receptors are not necessary to maintain mismatch and visual responses once V1 is fully trained by visuomotor experience.

Both a visuomotor mismatch and the sudden appearance of a visual stimulus are unpredictable events and can be interpreted as negative and positive prediction errors, respectively. Assuming there is indeed a deficit in the development of prediction error responses induced by the NMDA receptor knockout, we would also expect a deficit in the suppression of predictable responses. To investigate this, we quantified the suppression of running onset responses by visual flow in the closed-loop condition. In normally reared mice, a running onset with closed-loop visual feedback is typically associated with an increase in activity that is transient, whereas the response to the same running onset in darkness results in a sustained change in activity (*Figure 4A*). One interpretation of this is that the visual flow coupled to locomotion in the closed-loop condition triggers a suppression of the running-related responses. We quantified the suppression in the closed-loop condition by calculating the difference between the running onset response in darkness and that in the closed-loop condition (*Figure 4A*). Computing this difference for control mice, $\Delta Grin1_{juv}$ mice, and $\Delta Grin1_{adult}$ mice, we found that this suppression was absent only in the knockout hemisphere of the $\Delta Grin1_{juv}$ mice (*Figure 4B and C*). This is consistent with an impairment in the suppression of predictable responses in L2/3 neurons by an NMDA receptor knockout prior to first visual experience.

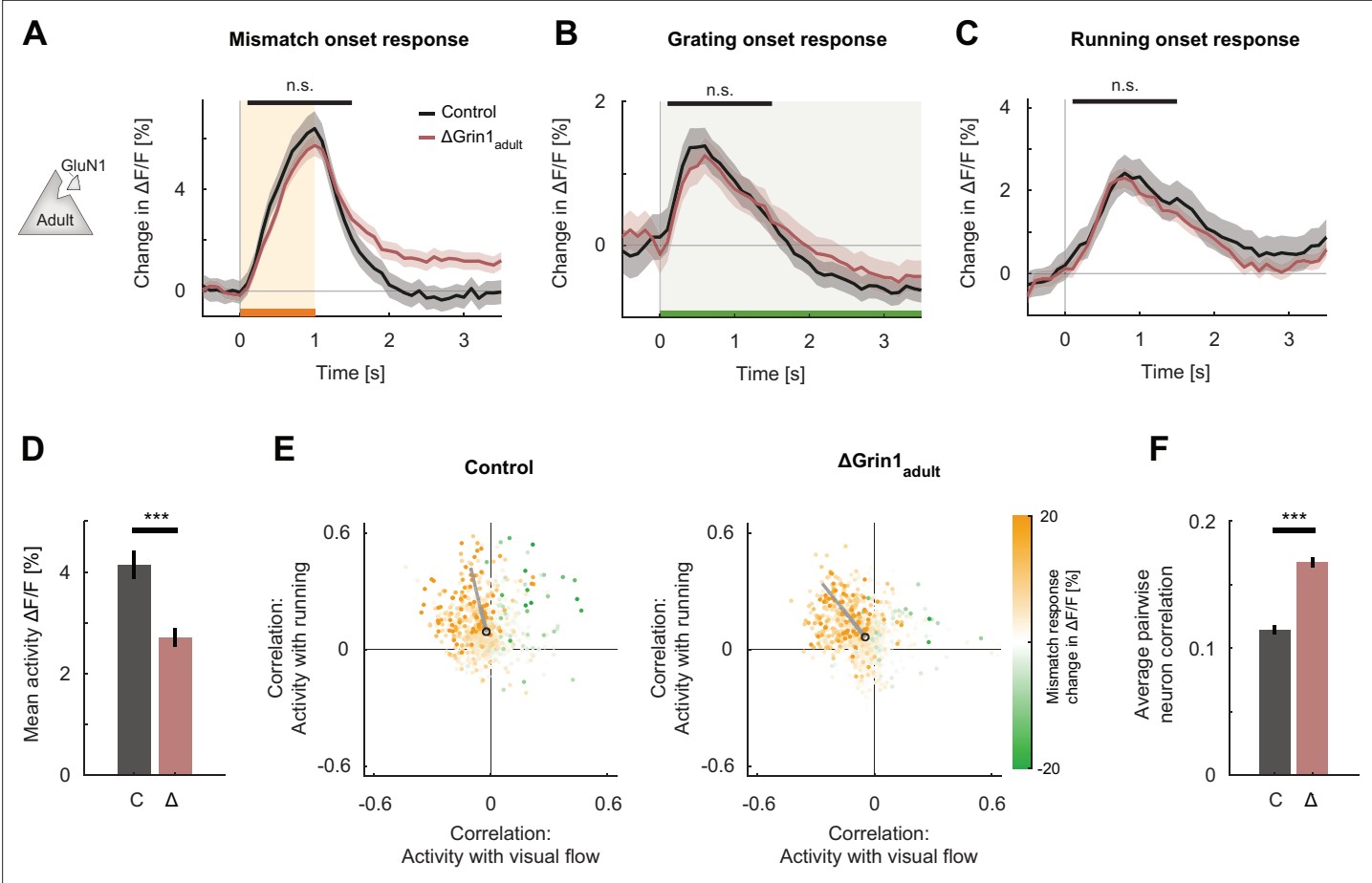

**Figure 3.** *N*-methyl-ᴅ-aspartate (NMDA) receptor knockout in the adult mouse did not impair visual and visuomotor responses. (**A**) The average population response to mismatch was similar in control (black) and in ΔGrin1_adult (dark red) hemispheres. Shading indicates the standard error of the mean (SEM) across neurons. Orange shading and bar indicate the duration of mismatch. Mean responses were compared across neurons in the time window indicated by the black bar above the traces. Here and in subsequent panels, n.s.: p>0.05, *p<0.05, **p<0.01, ***p<0.001. For all details of statistical testing, see ***Supplementary file 1A***. (**B**) As in (**A**), but for responses to the onset of a drifting grating stimulus (see Materials and methods). Green shading and bar indicate the presence of grating stimulus. (**C**) As in (**A**), but for running onset responses in the closed-loop condition. (**D**) Mean activity of neurons in the control (**C**, gray) and ΔGrin1_adult (Δ, dark red) hemisphere during the closed-loop condition. Error bars indicate SEM across neurons. (**E**) Scatter plot of the correlation between neuronal activity and visual flow, and the correlation between neuronal activity and running speed in the open-loop condition for all L2/3 neurons recorded in control (left) and ΔGrin1_adult (right) hemispheres. Each dot shows the correlations for one neuron, and dot color indicates the neuron's mismatch response. Black circles mark the population mean, and solid gray lines indicate the direction of the first principal component of the distribution (see Materials and methods). (**F**) Average pairwise correlation of neuronal activity was higher in ΔGrin1_adult (Δ, dark red) compared to that in the control (**C**, gray) hemisphere. Error bars indicate SEM across neurons.

## Local NMDA receptor dysfunction during development resulted in impaired visuomotor skill learning later in life

Assuming NMDA receptor-dependent signaling during development is necessary for the establishment of normal visuomotor integration, we expected that the ΔGrin1_juv mice would exhibit behavioral impairments in cortex-dependent visuomotor tasks. To test this, we trained six ΔGrin1_juv mice in a visuomotor task later in life. For these experiments, we used two control groups. The first was composed of 13 ΔGrin1_adult mice, and the second was composed of 6 control mice (Control_juv) that did not receive a *Grin1* knockout but were dark-reared from birth. The ΔGrin1_juv and Control_juv groups were dark-reared until P32. All three groups were initially exposed to closed-loop experience in a virtual reality setup (as described above) and subsequently trained to perform a virtual navigation task (***Heindorf et al., 2018***; ***Figure 5A and B***). In this task, mice had control over movement in a virtual environment through rotation and forward locomotion on a spherical treadmill and were trained to reach the end of a virtual corridor for a water reward. Training lasted for 7 days, 1 hr per day.

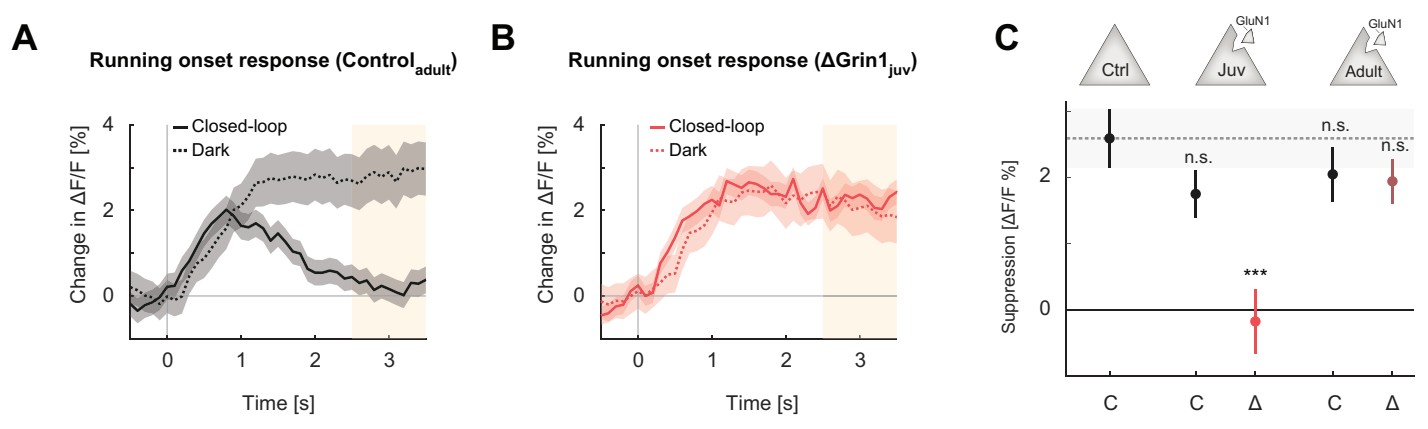

**Figure 4.** Suppression of running onset responses by visual flow was reduced by an *N*-methyl-ᴅ-aspartate (NMDA) receptor knockout prior to first visual experience. (**A**) The average population response to running onset in the closed-loop condition (solid) and the dark condition (dotted) in adult control mice. Shading indicates the standard error of the mean (SEM) across neurons. Albescent white shading marks analysis window used in (**C**). Note that the visual flow associated with closed-loop running results in a suppression of motor-related responses. (**B**) As in (**A**), but for $\Delta Grin1_{juv}$ data in the knockout hemisphere. (**C**) Average closed-loop visual feedback induced suppression of activity for all neurons in adult control mice and control (**C**) or knockout (Δ) hemispheres of $\Delta Grin1_{juv}$ and $\Delta Grin1_{adult}$ mice. Suppression was calculated as the difference between the running onset response in the dark and the closed-loop condition in the window 2.5–3.5 s after running onset, marked in (**A**) and (**B**). Error bars indicate SEM across neurons. Comparison against data from control mice; n.s.: p>0.05, ***p<0.001. For all details of statistical testing, see ***Supplementary file 1A***.

We quantified performance using an index that is based on the fraction of distance traveled toward the target, normalized by the total distance traveled (see Materials and methods). The dark-reared $Control_{juv}$ mice and the adult knockout $\Delta Grin1_{adult}$ mice both learned to perform the task over the course of the training. The $\Delta Grin1_{juv}$ mice, however, failed to show evidence of increased performance over the course of the 7 days of training and exhibited significantly reduced performance compared to the two control groups late in training (***Figure 5C***). To test for the mice's ability to induce a behavioral response to an unexpected perturbation of visual feedback, we introduced sudden offsets of the current heading direction at random times by 30° either to the left or to the right. Mice typically learn to respond to these perturbations with a turn that corrects for the offset. Both $Control_{juv}$ and $\Delta Grin1_{adult}$ mice reacted with a compensatory turn in the correct direction by the end of training (***Figure 5D***). The $\Delta Grin1_{juv}$ mice, however, failed to correct for these offsets. Quantifying this as the learning-related change in offset perturbation response, we found that $Control_{juv}$ and $\Delta Grin1_{adult}$ mice both exhibit larger learning-related changes than the $\Delta Grin1_{juv}$ mice (***Figure 5E***). Thus, consistent with the dependence of normal visuomotor integration on NMDA receptors during the first visuomotor experience, we found that mice that lack NMDA receptors during the first visuomotor experience are impaired in learning this cortex-dependent, visually guided motor task later in life.

## CaMKII-dependent signaling in SST interneurons was necessary for feed-forward visual inhibition

Central to the subtractive computation of prediction error responses are inhibitory interneurons. By implementing the opposing influence of visual and locomotion-related input in L2/3 neurons (***Jordan and Keller, 2020***), they allow for a subtraction of a bottom-up sensory input and a top-down prediction to compute prediction errors (***Keller and Mrsic-Flogel, 2018***). Based on measurements of calcium responses to visuomotor mismatches and artificial manipulations of activity in different interneuron subtypes, we have previously speculated that a subset of somatostatin (SST)-positive interneurons mediates the visually driven inhibition necessary for mismatch responses in V1 L2/3 excitatory neurons (***Attinger et al., 2017***). We thus set out to test whether an impairment of CaMKII-dependent signaling selectively in SST interneurons in V1 during the first visuomotor experience would result in a failure to establish visually driven inhibition in L2/3 excitatory neurons. To do this, we turned to a method that would allow us to target the intervention specifically to SST interneurons in V1. We used a paAIP2 (***Murakoshi et al., 2017***) to inhibit CaMKII using blue light illumination. Once activated, paAIP2 binds to CaMKII and inhibits its activity. Upon cessation of illumination, paAIP2 dissociates from CaMKII

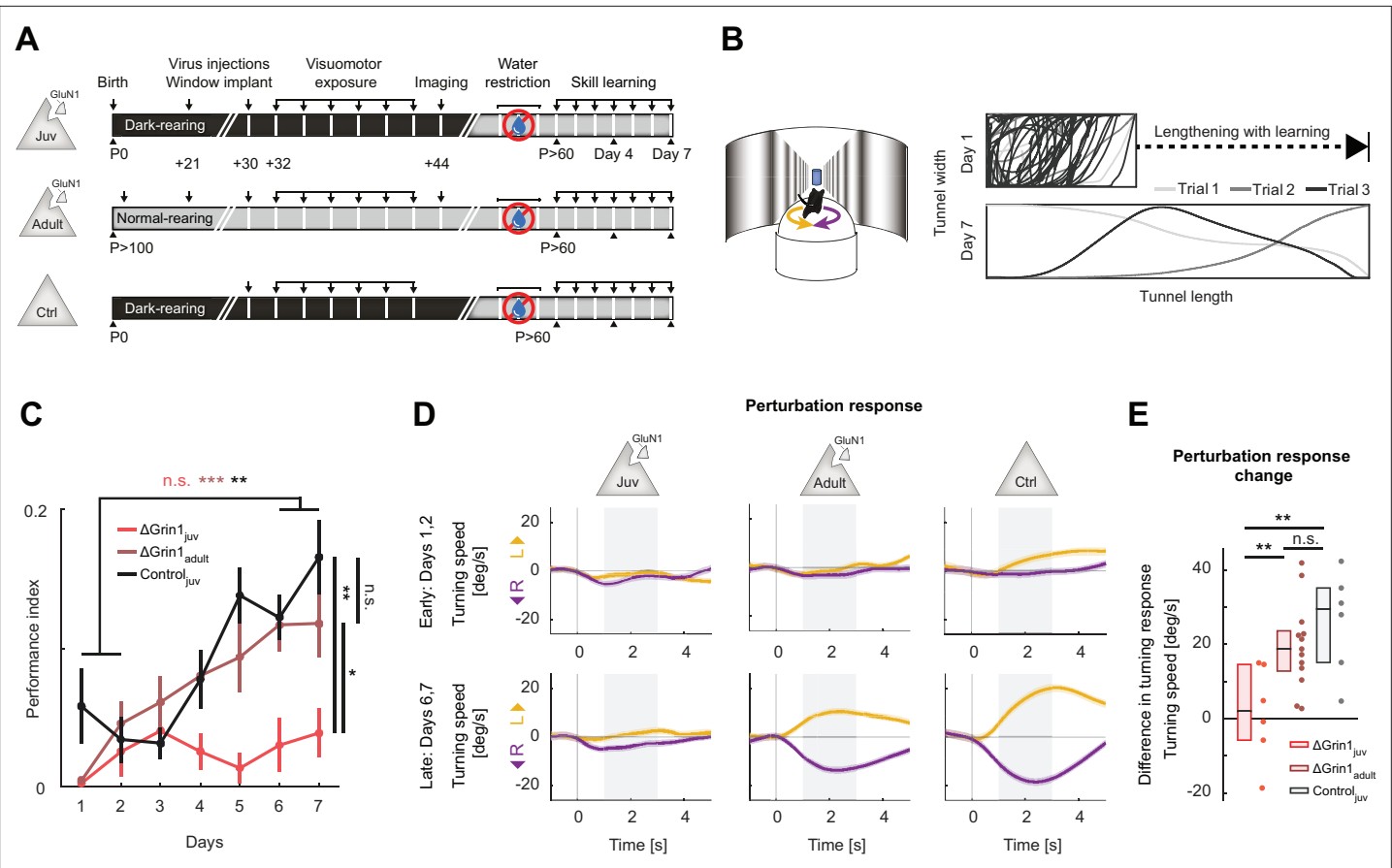

**Figure 5.** *N*-methyl-D-aspartate (NMDA) receptor knockout in V1 before first visuomotor experience impaired learning of a visuomotor task later in life. (**A**) Experimental approach and timeline. Three groups of mice were trained: the first was composed of 6 ΔGrin1_juv dark-reared mice, the second was composed of 13 ΔGrin1_adult normally reared mice, and the third was composed of 6 C57BL/6 dark-reared control mice. Mice were water-restricted and subsequently trained to perform a virtual navigation task (see Materials and methods). (**B**) Left: schematic of virtual reality setup. Mice controlled forward translational motion and rotation in a virtual corridor by rotating a spherical treadmill and were trained to navigate to the end of a corridor for a water reward. As performance increased, the task difficulty was increased by lengthening the virtual corridor. Right: top-down view of the virtual corridor showing the trajectories of the mouse in three example trials (different gray levels) on days 1 (top) and 7 (bottom). The ratio of virtual corridor length to width is not drawn to scale. (**C**) Task performance as a function of training day (see Materials and mthods) of ΔGrin1_juv mice (red), ΔGrin1_adult mice (dark red), and dark-reared control mice (Control_juv, black) over the course of 7 days. Error bars indicate the standard error of the mean (SEM) across mice. ΔGrin1_adult and Control_juv mice exhibited performance improvements over the course of training, while ΔGrin1_juv mice did not. Performance on day 7 was different between ΔGrin1_juv and both ΔGrin1_adult and Control_juv mice. Here and in subsequent panels, n.s.: p>0.05, *p<0.05, **p<0.01, ***p<0.001. For all details of statistical testing, see ***Supplementary file 1A***. (**D**) Turning in response to a perturbation that consisted of a sudden heading displacement of 30° to the left (yellow) or to the right (purple) of ΔGrin1_juv, ΔGrin1_adult, and Control_juv mice, early (top row) and late (bottom row) in training. Shading indicates SEM across trials. Gray shading indicates analysis window (+1 s to +3 s) used for quantification in (**E**). (**E**) Quantification of perturbation offset responses shown in (**D**) as the difference between average left and right perturbation turning responses, late (bottom row in **D**) minus early (top row in **D**) in training. Boxes show median and quartiles, all data are shown as dots (individual mice) to the right. ΔGrin1_adult and Control_juv mice learned to initiate corrective turns in response to visual offset perturbations, while ΔGrin1_juv mice did not.

over a time course of approximately 40 s. Thus, CaMKII inhibition can be controlled by the duration of illumination.

We repeated the experiments we performed with the NMDA receptor knockout using paAIP2 in three groups of mice to target CaMKII inhibition either to excitatory neurons, SST interneurons (***Figure 6A***), or parvalbumin (PV)-positive interneurons (***Figure 6—figure supplement 2A***). Again, all mice were dark-reared from birth and received 2 hr of visuomotor experience in the virtual reality environment every other day for 12 days (***Figure 6B***, ***Figure 6—figure supplement 2B***). The first group consisted of six C57BL/6 mice that received an injection of an AAV to express paAIP2 under a CaMKIIα(1.3 kb) promoter (AAV2/1-CaMKIIα-mEGFP-P2A-paAIP2) in excitatory neurons unilaterally in V1. The other two groups consisted of seven SST-Cre mice and six PV-Cre mice that each

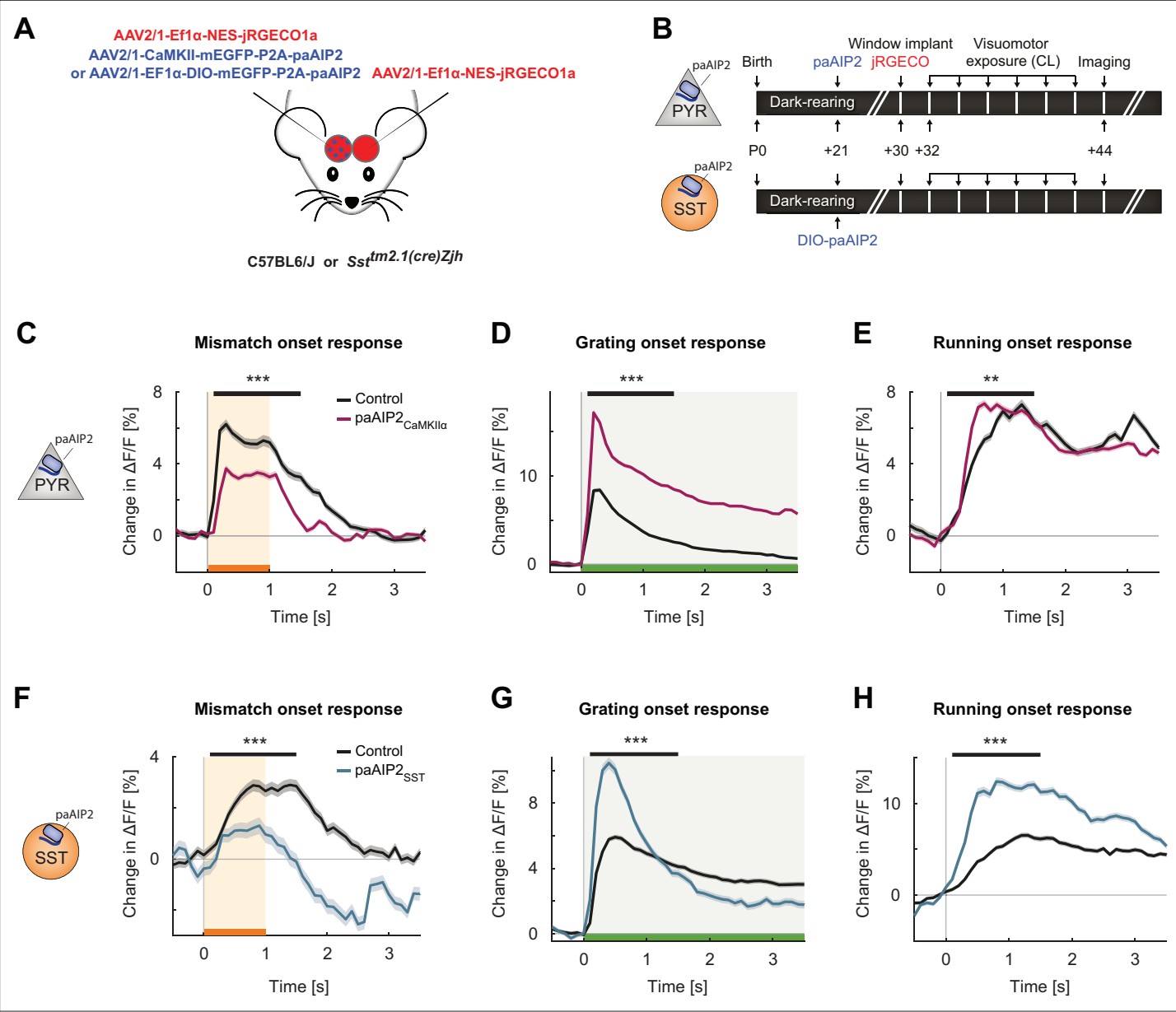

**Figure 6.** Inhibiting calcium/calmodulin-dependent kinase II (CaMKII) in excitatory neurons or somatostatin (SST) interneurons resulted in imbalanced visuomotor responses in L2/3 excitatory neurons. (**A**) We injected an AAV2/1-CaMKII-mEGFP-P2A-paAIP2 (in C57BL/6J mice) or AAV2/1-Ef1α-mEGFP-P2A-paAIP2 (in SST-Cre mice) unilaterally in V1 to express the photoactivatable CaMKII inhibitor paAIP2 in excitatory or SST interneurons, respectively, and an AAV2/1-Ef1α-NES-jRGECO1a bilaterally to express the calcium indicator jRGECO for imaging in L2/3 excitatory neurons. (**B**) Mice were dark-reared from birth. Adeno-associated viral vector (AAV) injections occurred at postnatal day 21 (paAIP2 or DIO-paAIP2) and P30 (jRGECO1a). Imaging window implantation occurred on P30. Mice had six sessions of visuomotor exposure in a closed-loop (CL) virtual environment during which we illuminated cortex bilaterally with blue light (473 nm) to inhibit CaMKII. We used six C57BL/6J mice, in which paAIP2 was targeted to excitatory neurons using a CaMKIIα(1.3 kb) promoter (paAIP2$_{CaMKIIα}$), and seven SST-Cre mice that received an injection of the DIO-paAIP2 vector (paAIP2$_{SST}$). (**C**) The average L2/3 population response to mismatch was stronger in control (black) than in paAIP2$_{CaMKIIα}$ (purple) hemispheres. Shading indicates the standard error of the mean (SEM) across neurons. Orange shading and bar indicate the duration of mismatch. Mean responses were compared across neurons in the time window marked by the black bar above the traces. Here and in subsequent panels, n.s.: $p > 0.05$, *$p < 0.05$, **$p < 0.01$, ***$p < 0.001$. For all details of statistical testing, see *Supplementary file 1A*. (**D**) As in (**C**), but for responses to the onset of a drifting grating stimulus (see Materials and methods). Green shading and bar indicate the presence of grating stimulus. (**E**) As in (**C**), but for running onset responses in the CL condition. (**F**) As in (**C**), but for inhibition of CaMKII in SST interneurons. (**G**) As in (**D**), but for inhibition of CaMKII in SST interneurons. (**H**) As in (**E**), but for inhibition of CaMKII in SST interneurons.

The online version of this article includes the following figure supplement(s) for figure 6:

*Figure 6 continued on next page*

*Figure 6 continued*

**Figure supplement 1.** Additional data for calcium/calmodulin-dependent kinase II (CaMKII) inhibition in excitatory or somatostatin (SST) inhibitory neurons.

**Figure supplement 2.** Inhibiting calcium/calmodulin-dependent kinase II (CaMKII) in parvalbumin (PV) interneurons resulted in an overall increase in onset responses in L2/3 excitatory neurons.

**Figure supplement 3.** Changes induced by calcium/calmodulin-dependent kinase II (CaMKII) inhibition quickly reverted with exposure to normal visuomotor coupling.

received an injection of AAV2/1-Ef1α-DIO-mEGFP-P2A-paAIP2 unilaterally in V1. At P30, prior to first visuomotor experience, mice were injected with an AAV to express a red-shifted calcium indicator (AAV2/1-Ef1α-NES-jRGECO1a) in both visual cortices to enable calcium imaging of L2/3 excitatory neurons. To activate paAIP2 throughout visuomotor exposure while mice were on the virtual reality setup, we illuminated V1 bilaterally with a blue (473 nm) laser using a 0.2 Hz stimulation protocol (see Materials and methods) for the entire duration of visuomotor exposure. As before (***Figures 2 and 3***), we then proceeded to measure mismatch, grating, and running onset responses in L2/3 excitatory neurons at P44. During these imaging experiments, paAIP2 was not activated. Similar to the responses observed in ΔGrin1$_{juv}$ mice, we found that in mice that expressed paAIP2 in excitatory neurons under the CaMKIIα(1.3 kb) promoter, the strongest changes were in mismatch and visual responses, while running onset responses were less affected (***Figure 6C–E***). Mismatch responses were again reduced in the inhibited hemisphere compared to the control hemisphere (***Figure 6C***). Intriguingly, CaMKII inhibition resulted in a massive increase in visually driven activity of L2/3 neurons (***Figure 6D***), as opposed to the reduction of visual responses we observed with the NMDA receptor knockout (***Figure 2***). We speculate that this difference can be explained by the fact that the power of light used to activate paAIP2 falls off exponentially with cortical depth (***Figure 6—figure supplement***

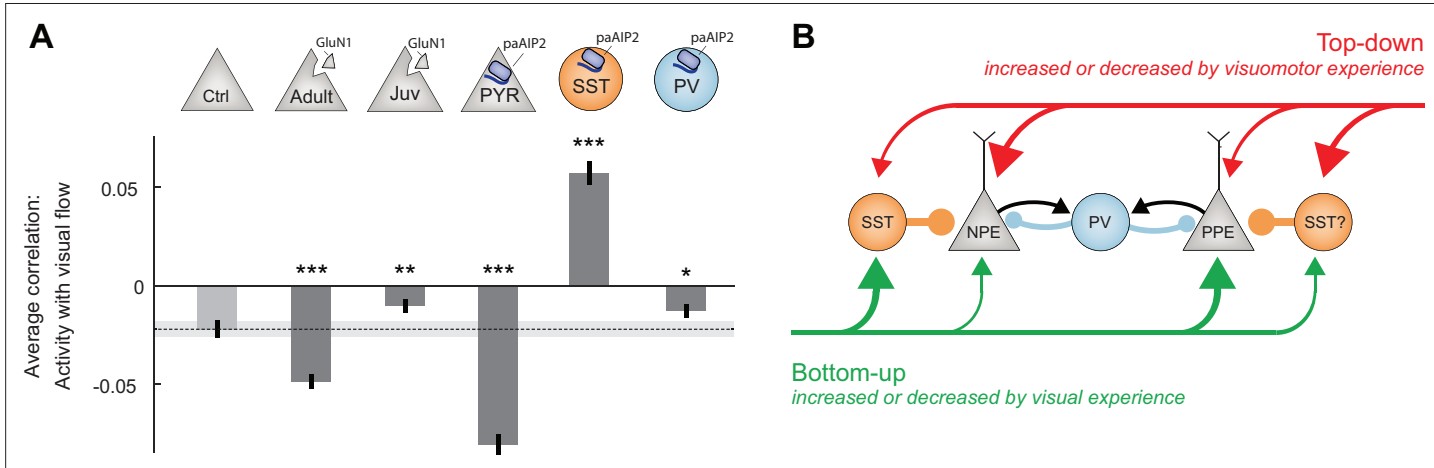

**Figure 7.** Calcium/calmodulin-dependent kinase II (CaMKII) inhibition in somatostatin (SST) interneurons during the first visuomotor experience reduced visually driven inhibition. (**A**) Mean correlation between neuronal activity and visual flow in the open-loop condition for all L2/3 excitatory neurons recorded in adult control, ΔGrin1$_{adult}$, ΔGrin1$_{juv}$, paAIP2$_{CaMKIIα}$, paAIP2$_{SST}$, and paAIP2$_{PV}$ mice. Error bars indicate the standard error of the mean (SEM) across neurons. Dashed black line and corresponding gray shading indicate the mean correlation of activity and visual flow and SEM of the adult control group; gray shading indicates SEM across neurons. Comparison against adult control data: n.s.: p>0.05, **p<0.01, ***p<0.001. For all details of statistical testing, see ***Supplementary file 1A***. (**B**) Through visuomotor experience, local plasticity in V1 establishes a balance between top-down and bottom-up input in L2/3 neurons (***Jordan and Keller, 2020***), which is thought to drive prediction error responses. In this model, we refer to neurons that receive strong bottom-up excitation and strong top-down inhibition as positive prediction error (PPE) neurons, while those that receive strong top-down excitation and strong bottom-up inhibition, we refer to as negative prediction error (NPE) neurons. Given that interfering with plasticity in either excitatory neurons or SST interneurons prevents normal development of visual responses in excitatory neurons, combined with the finding that visual responses in neither population of neurons depend on coupled visuomotor experience (***Attinger et al., 2017***), we conclude that visual experience is necessary and sufficient for shaping visual inputs onto both populations of neurons. As mismatch responses in excitatory neurons depend on visuomotor experience and are sensitive to blocking plasticity in excitatory neurons, the proper wiring of top-down input onto L2/3 excitatory neurons likely requires coupled visuomotor experience. SST interneurons likely mediate visually driven inhibition, and we speculate that they also mediate the top-down motor-related inhibition. The effect of interfering with plasticity in parvalbumin (PV) interneurons is consistent with the idea that they regulate overall gain of the circuit.

*1A*). Thus, CaMKII inhibition likely predominantly influenced superficial synapses, which preferentially carry top-down signals (see Discussion). Despite the difference in the effect of the manipulation on visually driven responses, we found an increase in correlations between the activity of L2/3 neurons on CaMKII inhibition, similar to that observed in the ΔGrin1$_{juv}$ mice (*Figure 6—figure supplement 1B*).

Given the differences between ΔGrin1$_{juv}$ and the CaMKII inhibition in excitatory neurons, we compared the effect of CaMKII inhibition in inhibitory interneurons to that observed when inhibiting CaMKII in excitatory neurons. Inhibiting CaMKII in SST interneurons had an effect similar to that of inhibiting CaMKII in excitatory neurons, decreasing mismatch responses and increasing visual responses (*Figure 6F and G*). Interestingly, the running onset responses during the closed-loop condition were much larger in the inhibited hemisphere (*Figure 6H*). This could be explained by an increased motor-related excitatory input, a decreased bottom-up visual inhibition, or a combination of both. Assuming SST interneurons mediate visually driven inhibition, and that the establishment of this inhibition is experience dependent (*Attinger et al., 2017*), we would expect CaMKII inhibition in SST interneurons to result in decreased visually driven inhibition onto L2/3 neurons. To test this, we quantified the average correlation between neuronal activity and visual flow speed in open-loop condition. Under normal conditions, this correlation is negative for L2/3 excitatory neurons (*Figure 7A*). The correlation became more strongly negative with the inhibition of CaMKII in excitatory neurons but became positive with the inhibition of CaMKII in SST interneurons. This is consistent with a decrease in visually driven inhibition by the paAIP2 inhibition of CaMKII in SST interneurons. To test whether this effect is specific to SST interneurons or simply the consequence of altering inhibition, we repeated these experiments in mice that expressed paAIP2 in PV interneurons. Consistent with a role of PV interneurons in modulating cortical gain (*Atallah et al., 2012*), inhibiting CaMKII in PV interneurons resulted in a uniform increase in all response types (*Figure 6—figure supplement 2C–E*), but did not lead to a net positive correlation of neuronal activity with visual flow in excitatory neurons (*Figure 7A*). Moreover, comparing the average correlation of activity and visual flow across all experimental manipulations, we found that only the inhibition of CaMKII in SST interneurons resulted in a net positive correlation of neural activity with visual flow in L2/3 excitatory neurons. Thus, plasticity in SST interneurons is likely central to establishing normal levels of visually driven inhibition in V1.

To test if normal visuomotor experience without inhibition of CaMKII would revert the changes we observed, we returned the mice to dark housing for 2 days following the first imaging session and repeated the neuronal activity measurements. At the time of the second measurement, the only visual experience without inhibition of CaMKII the mice had was approximately 15 min of closed-loop visual feedback, 30 min of open-loop visual flow, and 15 min of grating stimuli in the first imaging session. We found that after this 1 hr of visual experience, most of the CaMKII inhibition-induced effects had either significantly reduced or reverted. For mice with inhibition of CaMKII in excitatory neurons or SST interneurons, mismatch responses in the inhibited hemisphere were larger on the second day of imaging than on the first day of imaging (*Figure 6—figure supplement 3A and D*), while grating onset responses were significantly reduced compared to the first day of imaging (*Figure 6—figure supplement 3B and E*). Running onset responses in the closed-loop condition on the second day of imaging were decreased in the inhibited hemisphere compared to those on the first day of imaging (*Figure 6—figure supplement 3C and F*), and the correlation of neuronal activity with visual flow became negative in the mice that had originally received CaMKII inhibition in SST interneurons (*Figure 6—figure supplement 3G*). Thus, normal visuomotor coupling in the absence of CaMKII inhibition induced changes that were directly opposing those induced by the CaMKII inhibition. Together, these data are consistent with the interpretation that plasticity both in the top-down input to L2/3 as well as the visually driven inhibition mediated by SST interneurons is necessary to establish the L2/3 circuit underlying the computation of visuomotor prediction errors.

## Discussion

Our results demonstrate that with first visual experience in the life of a mouse, exposure to visuomotor coupling establishes a circuit in V1 capable of integrating motor and visual signals that is necessary for the acquisition of certain visuomotor skills later in life. We found that interfering with a subset of the plasticity mechanisms in visual cortex, through a local knockout of NMDA receptors, impaired the establishment of this circuit. The NMDA receptor knockout resulted in a reduction of responses in L2/3 neurons to mismatch and visual stimuli (*Figure 2*) that can be interpreted as a reduced capacity

of V1 to compute visuomotor prediction errors. More specifically, considering that L2/3 excitatory neurons balance opposing bottom-up and top-down input (*Jordan and Keller, 2020*), our results indicate that this balance is established by local plasticity in V1 through experience with visuomotor coupling early in life. Moreover, given that the same manipulation also impaired the ability of mice to learn a visuomotor task later in life, we speculate that the ability of V1 to compute visuomotor prediction errors is an essential component of the computational strategy the brain uses to guide movement by visual feedback in complex behavioral tasks. Interestingly, later in life, NMDA receptors in V1 are no longer necessary for visuomotor skill learning, indicating that in this case most of the learning-related plasticity occurs outside of V1, or independent of NMDA receptors.

When interpreting our results, it should be kept in mind that our strategy to knock out NMDA receptors in V1 is not specific to L2/3 neurons, and we cannot be certain if the effects we observed in L2/3 neurons are the direct consequence of the NMDA receptor knockout in these neurons or a downstream consequence of an effect in another layer. A balance of opposing bottom-up and top-down input could be established either by matching a bottom-up input to a fixed top-down input, or vice versa, or by changing both bottom-up and top-down input onto excitatory L2/3 neurons. Alternatively, it is possible that there is a reduction of bottom-up input onto L4 neurons, and hence reduced bottom-up input onto L2/3 neurons. We think this is unlikely as a knockout of NMDA receptors in L4 neurons, the main source of bottom-up visual input to L2/3 neurons, does not alter visually evoked potentials in visual cortex, nor does it impair visual acuity of the mice, regardless of whether the knockout is congenital or postadolescent (*Fong et al., 2020*; *Sawtell et al., 2003*). Thus, we speculate that the NMDA receptor knockout effects we observed are at least in part driven by interfering with the establishment of normal input to the L2/3 neurons in V1. Another potential confound of these experiments is that we were using intracellular calcium concentration changes to measure neuronal activity, when the NMDA receptor channel is permeable to calcium and constitutes the main source of calcium in dendritic spines (*Sabatini et al., 2002*). However, given that we are measuring calcium signals at the soma where the main source of calcium is voltage-gated calcium channels (*Grienberger and Konnerth, 2012*), the direct effect of the NMDA receptor knockout on intracellular calcium is unlikely to interfere with our conclusions. Moreover, an overall reduction in calcium would influence all responses equally and would not explain why after NMDA receptor knockout we found a strong reduction in mismatch and visual responses but only a small reduction in mean activity levels in $\Delta Grin1_{juv}$ mice (*Figure 2*), while in in $\Delta Grin1_{adult}$ mice the converse was true (*Figure 3*).

There was a marked difference between the NMDA receptor knockout and the CaMKII inhibition, in that the latter led to a massive increase in visual responses. There are several possible explanations that could account for this difference. First, even though NMDA receptors and CaMKII are closely linked in many forms of synaptic plasticity, there could be a systematic difference in the dependence of plasticity on the two molecules as a function of neuron or synapse type. Second, while the NMDA receptor knockout was permanent, we only inhibited CaMKII during the visuomotor training. Outside of this time, when the mice were housed in darkness, there could have been forms of compensatory CaMKII-dependent plasticity in V1 in response to either visuomotor experience-driven plasticity outside of V1 or non CaMKII-dependent plasticity in V1. Third, while the NMDA knockout affected all neuron types similarly (*Figure 1G*), the CaMKII inhibition predominantly targeted subsets of excitatory neurons, SST-positive interneurons, or PV-positive interneurons. Interestingly, however, with respect to the increase in visual responses, CaMKII inhibition in any one of these three neuron types resulted in an increase of visual responses in L2/3 excitatory neurons (*Figure 6D and G*, *Figure 6—figure supplement 2D*). Thus, differences in the neuron types targeted are unlikely to explain the different effect of NMDA receptor knockout or CaMKII inhibition on visual responses. Fourth, as the inhibition of CaMKII is driven by blue light illumination on the cortical surface, there could be a systematic difference in which synapses, or neurons, were influenced by the manipulation. The power of the light used to activate paAIP2 falls off exponentially with cortical depth with an estimated decay constant of less than 100 μm (*Figure 6—figure supplement 1A*; *Yona et al., 2016*). This, combined with the fact that CaMKII expression is higher in superficial L2/3 neurons than L4 or L5 neurons (*Lein et al., 2007*; *Tighilet et al., 1998*), could result in an increased effect of the CaMKII inhibition in superficial synapses. Long-range cortical input, which is thought to carry motor-related input to V1 (*Leinweber et al., 2017*), arrives preferentially on more superficial synapses than the bottom-up visual input (*Park et al., 2019*; *Petreanu et al., 2009*; *Young et al., 2021*). Thus, the differences in effect on grating

responses between the NMDA receptor knockout and the CaMKII inhibition could be explained by a differential influence on top-down and bottom-up pathways. While we cannot exclude the involvement of the other potential explanations discussed above, it is not immediately clear why they would result in a differential effect with regards to positive and negative prediction errors.

Population responses to mismatch stimuli, but also to grating stimuli and running onsets, vary considerably across mice. One factor that influences this variability is differences in rearing conditions. Dark rearing is known to delay normal development of V1 (*Hensch, 2005*; *Sherman and Spear, 1982*). ΔGrin1$_{juv}$ mice, for example, were dark-reared while the ΔGrin1$_{adult}$ mice were normally reared. This dark rearing of the ΔGrin1$_{juv}$ mice was necessary to enable a local NMDA knockout in V1 prior to first visuomotor experience. We know from previous work, however, that this paradigm of dark rearing followed by visuomotor training in a virtual environment does not impair normal development of visuomotor integration (*Attinger et al., 2017*), and hence should not influence our conclusions. Nevertheless, to minimize the influence of variability across mice, we used an experimental design that is based on a within-animal control hemisphere that was not manipulated. It is important to note, however, that the within-animal control suffers from the confound that the two hemispheres are directly connected. For instance, the fact that visual responses were also massively increased in the control hemisphere of CaMKII-inhibited mice compared to the level of responses one would expect normally (e.g., compare *Figure 6D* with *Figure 3B*, or *Attinger et al., 2017*) is likely caused by this direct interaction. And similarly for the fact that upon exposure to normal visuomotor in the absence of CaMKII inhibition the reversal of responses in the inhibited hemisphere often overshot those observed in the control hemisphere at first measurement (*Figure 6—figure supplement 3*). In the case of mismatch responses in the experiments in which we inhibited CaMKII in excitatory neurons (*Figure 6C*), or PV-positive interneurons (*Figure 6—figure supplement 2C*), the responses in the control hemisphere were larger and smaller, respectively, than what we would expect from control mice. This could be the result of a neuronal circuit strategy to maintain mismatch response at a constant level (*Liebscher et al., 2016*). A similar problem befalls our experiments using the NMDA receptor knockout. However, given that the effect sizes were considerably smaller in those experiments, crosstalk effects were likely also less salient. Thus, while there are caveats to the within-animal control, these should not alter our conclusions.

Lastly, given that little is known about the role of CaMKII in the plasticity in interneurons, it was not a priori clear that blocking CaMKII in SST or PV interneurons during visuomotor development would have a measurable influence on L2/3 excitatory neuron responses. While CaMKIIα is mainly expressed in excitatory neurons in cortex, CaMKIIβ is found in both excitatory and inhibitory neurons (*Nicole and Pacary, 2020*). Given that paAIP2 is designed based on a sequence of the autoinhibitory domain of CaMKII (*Hanson et al., 1989*) that is highly conserved across isoforms (*Tobimatsu and Fujisawa, 1989*) and inhibits CaMKII at the kinase domain (*Murakoshi et al., 2017*), which is also highly conserved across isoforms (*Tobimatsu and Fujisawa, 1989*), paAIP2 inhibition is likely independent of CaMKII isoform. Thus, our results would be consistent with the interpretation that SST and PV interneurons exhibit CaMKIIβ-dependent forms of plasticity necessary for the establishment of normal visuomotor integration in V1. Supporting this interpretation is the fact that inhibiting CaMKII in SST interneurons had an effect on net visual drive opposite to that of the same inhibition in excitatory neurons (*Figure 7A*). Consistent with the previous finding that SST activity is critical for the computation of visuomotor mismatch responses (*Attinger et al., 2017*), the role of SST interneurons appears to be important to establishing a balance between top-down and bottom-up input in L2/3 neurons in V1. This is in line with the findings that the activity of SST interneurons is modulated by locomotion only in the presence of visual input (*Pakan et al., 2016*), and that in excitatory neurons, inputs from both excitatory neurons and SST interneurons, but not PV or VIP-expressing interneurons, exhibit NMDA receptor-dependent plasticity (*Chiu et al., 2018*). Thus, we postulate that during visual development visuomotor experience establishes a balance in individual L2/3 neurons, either between a top-down excitatory input and a visually driven inhibitory input mediated by SST interneurons, or a top-down inhibitory input – possibly also mediated by SST interneurons – and a visually driven excitatory input (*Figure 7B*).

## Materials and methods

**Key resources table**

| Reagent type (species) or resource | Designation | Source or reference | Identifiers | Additional information |
|---|---|---|---|---|
| Strain, strain background (*AAV*) | AAV2/1-EF1α-GCaMP6f-WPRE | FMI vector core | | $6.0 \times 10^{11}$–$8.0 \times 10^{12}$ GC/ml |
| Strain, strain background (*AAV*) | AAV2/1-EF1α-Cre-t2a-mcherry-WPRE | FMI vector core | | $3.2 \times 10^{11}$–$1.2 \times 10^{13}$ GC/ml |
| Strain, strain background (*AAV*) | AAV2/1-EF1α-Cre-WPRE | FMI vector core | | $2.8 \times 10^{10}$ GC/ml |
| Strain, strain background (*AAV*) | AAV2/1-EF1α-NES-jRGECO1a-WPRE | FMI vector core | | $4.8 \times 10^{13}$ GC/ml |
| Strain, strain background (*AAV*) | AAV2/1-CaMKIIα(1.3 kb)-mEGFP-P2A-paAIP2 | FMI vector core | | $1.80 \times 10^{13}$ GC/ml |
| Strain, strain background (*AAV*) | AAV2/1-EF1α-DIO-mEGFP-P2A-paAIP2-WPRE | FMI vector core | | $1.2 \times 10^{13}$ GC/ml |
| Strain, strain background (*AAV*) | AAV2/1-EF1α-mCherry-IRES-Flpo | FMI vector core | | $1.3 \times 10^{13}$ GC/ml |
| Strain, strain background (*Mus musculus*) | C57BL6/J | Charles River | | |
| Strain, strain background (*M. musculus*) | *Grin1*tm2Stl/J | Jackson Laboratories | Cat# 005246 | |
| Strain, strain background (*M. musculus*) | *Pvalb*tm1(cre)Arbr | Jackson Laboratories | Cat# 008069 | |
| Strain, strain background (*M. musculus*) | *Sst*tm2.1(cre)Zjh | Jackson Laboratories | Cat# 018973 | |
| Chemical compound, drug | Fentanyl citrate | Actavis | CAS 990-73-8 | |
| Chemical compound, drug | Midazolam (Dormicum) | Roche | CAS 59467-96-8 | |
| Chemical compound, drug | Medetomidine (Domitor) | Orion Pharma | CAS 86347-14-0 | |
| Chemical compound, drug | Ropivacaine | Presenius Kabi | CAS 132112-35-7 | |
| Chemical compound, drug | Lidocaine | Bichsel | CAS 137-58-6 | |
| Chemical compound, drug | Buprenorphine | Reckitt Benckiser Healthcare | CAS 52485-79-7 | |
| Chemical compound, drug | Ophthalmic gel (Humigel) | Virbac | | |
| Chemical compound, drug | Flumazenil (Anexate) | Roche | CAS 78755-81-4 | |
| Chemical compound, drug | Atipamezole (Antisedan) | Orion Pharma | CAS 104054-27-5 | |
| Chemical compound, drug | N-Butyl-2-cyanoacrylate (Histoacryl) | Braun | CAS 6606-65-1 | |
| Chemical compound, drug | Dental cement (Paladur) | Heraeus Kulzer | CAS 9066-86-8 | |
| Chemical compound, drug | MK-801 | Sigma | CAS 77086-22-7 | |
| Software, algorithm | MATLAB (2020b) | MathWorks | RRID:SCR_001622 | |
| Software, algorithm | LabVIEW | National Instruments | RRID:SCR_014325 | |
| Software, algorithm | Two-photon acquisition software | Keller laboratory | https://sourceforge.net/p/iris-scanning/ | |
| Software, algorithm | Image data processing software | Keller laboratory | https://sourceforge.net/p/iris-scanning/calliope | |
| Software, algorithm | Python | https://python.org | RRID:SCR_008394 | |
| Software, algorithm | Panda3D | https://panda3d.org | | |
| Software, algorithm | R (v 4.0) | https://r-project.org | RRID:SCR_001905 | |
| Software, algorithm | LIGER (v 0.5) | (**Welch et al., 2019**) https://github.com/welch-lab/liger | RRID:SCR_018100 | |

| Reagent type (species) or resource | Designation | Source or reference | Identifiers | Additional information |
|---|---|---|---|---|
| Software, algorithm | DropletUtils (v 1.8) | https://bioconductor.org | | |
| Software, algorithm | Seurat (v 3.1.5.9008) | https://satijalab.org | RRID:SCR_016341 | |
| Software, algorithm | Scater (v 1.16.0) | https://bioconductor.org | RRID:SCR_015954 | |
| Other | mRNA probe Mm-Grin1-O1 (probe region 2892–4127) | ACD bio | Cat# 473079 | |
| Other | Optogenetic-stimulation laser (OBIS 473 nm LX) | Coherent | Cat# 1187194 | |

## Animals and surgery

All animal procedures were approved by and carried out in accordance with the guidelines of the Veterinary Department of the Canton Basel-Stadt, Switzerland, under license number 2573. For all surgical procedures, mice were anesthetized with a mixture of fentanyl (0.05 mg/kg; Actavis), midazolam (5.0 mg/kg; Dormicum, Roche), and medetomidine (0.5 mg/kg; Domitor, Orion). Analgesics were applied perioperatively (2% lidocaine gel, meloxicam 5 mg/kg) and postoperatively (buprenorphine 0.1 mg/kg, Metacam 5 mg/kg). Eyes were covered with ophthalmic gel (Virbac Schweiz AG). At P21, we injected approximately 100 nl of AAV2/1-Ef1α-Cre-T2A-mCherry vector at a titer of between $3.2 \times 10^{11}$ and $1.2 \times 10^{13}$ GC/ml, or AAV2/1-EF1α-Cre-WPRE vector at a titer of $2.8 \times 10^{10}$ GC/ml (*Figures 1–5*); AAV2/1-CaMKIIα(1.3 kb)-mEGFP-P2A-paAIP2 vector at a titer of $1.8 \times 10^{13}$ GC/ml, or AAV2/1-EF1α-DIO-mEGFP-P2A-paAIP2-WPRE vector at a titer of $1.2 \times 10^{13}$ GC/ml (*Figure 6*, *Figure 6—figure supplement 2*) through a small hole in the skull made over the right hemisphere at 2.4 mm lateral from lambda.

For window implantations at P30, we performed a cranial window surgery by implanting a circular 4 mm glass coverslip bilaterally, following injections of approximately 200 nl of AAV vectors (AAV2/1-EF1α-GCaMP6f-WPRE or AAV2/1-EF1α-NES-jRGECO1a-WPRE) into V1, 2.5 mm lateral from lambda.

## Virtual reality environment and virtual navigation task

In all experiments involving the virtual reality system, mice were head-fixed and mounted on a spherical treadmill, as described previously (*Leinweber et al., 2014*). In brief, mice were free to run on an air-supported polystyrene ball. Ball rotation controlled movement in a virtual reality environment displayed on a toroidal screen surrounding the mouse, which covered approximately 240° horizontally and 100° vertically of visual space, from the point of view of the mouse.

First visual and visuomotor exposure of the mice occurred in this virtual reality environment in the 12 days prior to imaging experiments. Mice were trained for 2 hr every other day (for a total of six sessions) with closed-loop feedback between forward locomotion and backward visual flow in a virtual corridor with walls textured with vertical sinusoidal gratings (*Attinger et al., 2017*). All two-photon imaging experiments were also performed on the same virtual reality setup, and data were acquired in sessions of 5–15 min duration in the following sequence: closed-loop, open-loop, dark, grating. In the closed-loop session, running was coupled to movement in the same virtual environment used during visuomotor exposure. In the open-loop session, self-generated visual flow from the preceding closed-loop session was replayed. In the grating sessions, drifting grating stimuli of different directions (0, 45, 90, 270°, moving in either direction) were presented in random sequences. Each grating presentation lasted between 3 s and 8 s with an inter-trial interval (gray screen) of between 2 s and 6 s. For all experiments, rotation of the ball was restricted to the transverse axis to allow only forward and backward movement in the virtual reality environment. Mice were free to run in all experiments and did so spontaneously.

For the virtual navigation experiments (*Figure 5*), rotation of the ball was not restricted, and mice could control forward and backward motion, as well as rotation in the virtual environment. To incentivize mice to engage in the visuomotor skill learning task, they were water-restricted with access to 1 ml water daily for 3 days before the start of the behavioral experiments. Care was taken to prevent a drop in body weight to below 80% of starting weight throughout training. During the experiment, mice could obtain water rewards by reaching the end of the virtual corridor, after which they were

presented with a 5 s gray screen and teleported to the beginning of the corridor. Task difficulty was increased with increasing performance of the mice by expanding the length of the virtual corridor to keep the rate of water rewards roughly constant. At the beginning of training, the length-to-width ratio of the corridor was 5. Every four trials, the length of the corridor would be updated by a factor between 1 (no change) and 1.5 (50% increase in length), where the factor was determined as 20 s divided by the mean duration of those four trials. Maximum corridor length was restricted to 400% of the length on the first day. Visual offset perturbations were introduced once per trial, presented at a random position within 20 and 80% of the total corridor length and consisted of 30° heading offsets introduced randomly, either to the left or to the right. The task performance index (PI) was calculated as follows:

$$PI = \frac{\int \cos\left(\theta\left(t\right)\right) * v\left(t\right) \, dt}{\int v\left(t\right) \, dt} * \frac{time \; spent \; running}{total \; time}$$

where $\theta(t)$ is the direction of running relative to the target, and $v(t)$ is the running speed of the mouse. The intuition behind this index is to quantify performance as the fraction of distance traveled in the direction of the target, normalized by the total distance traveled. The second factor is added to reduce variability driven by a short time spent running, as is typical in early training sessions.

## Two-photon calcium imaging

Two-photon imaging of L2/3 neurons in V1 was performed as described previously (*Leinweber et al., 2014*; *Leinweber et al., 2017*). In brief, two-photon imaging was performed using a modified Thorlabs Bergamo I or II microscope. The excitation light source was a tunable, femtosecond-pulsed laser (Insight, Spectra Physics, tuned to 910 nm or 980 nm for GCaMP6f excitation, and 1030 nm for jRGECO1a excitation). The scan head was based on an 8 kHz or 12 kHz resonant scanner (Cambridge Technology). We used a piezo electric linear actuator (P-726, Physik Instrumente) to sequentially image 4z-planes (approximately 40 μm apart) by moving a ×16, 0.8 NA objective (Nikon N16XLWD-PF). Emission light was band-pass filtered using a 525/50 nm or a 607/70 nm filter (Semrock), detected by a photomultiplier tube (PMT, H7422P, Hamamatsu), amplified (DHPCA-100, Femto), digitized at 800 MHz (7965R, National Instruments), and band-pass filtered at 80 MHz using digital Fourier transform on a field-programmable gate array (NI5772, National Instruments, loaded with custom-designed logic). Images were acquired at 750 × 400 pixels using custom-written LabVIEW software (available on a public SourceForge repository, see Key resources table), at 10 Hz or 15 Hz effective frame rate and a field of view of approximately 375 μm × 300 μm. Whenever possible, imaging was performed in both control and intervention hemisphere in each mouse. In a subset of mice (see *Supplementary file 1B*), imaging was only possible in one hemisphere as imaging quality did not meet our minimum quality standards (clear image visible in single frame at less than 60 mW total laser power) in the other hemisphere.

## Conditional *Grin1* knockout, histology, and pharmacological NMDA receptor inhibition

All ΔGrin1 knockout experiments were performed using the *Grin1^tm2Stl* mouse line (*Tsien et al., 1996*), which has a pair of loxP sites flanking the transmembrane domain and C-terminal region of the *Grin1* gene that codes for GluN1, a subunit essential to the NMDA receptor function (*Monyer et al., 1994*). We confirmed the knockout using mRNA in situ hybridization (RNAscope, Ventana) in a separate cohort of two mice (one P21, and one >P100), 14 days after injection of an AAV vector expressing Cre recombinase in both juvenile and adult mice (ΔGrin1_{juv, adult}). We followed a standardized formaldehyde-fixed paraffin-embedding protocol. In brief, mice were transcardially perfused with phosphate buffered saline (PBS), followed by perfusion with a solution of 4% paraformaldehyde (PFA) in PBS. Brains were isolated, post-fixed overnight in 4% PFA, paraffinized for 24 hr, and cut at 5 μm using a microtome (Thermo Fisher). Slices were stained using hematoxylin to mark cell bodies, and Mm-*Grin1*-O1 (#473079, target region 2892–4127, ACDBio) to label *Grin1* mRNA. To ease identification of the knockout area in two-photon calcium imaging, a vector co-expressing a red fluorophore (mCherry) and Cre was used to induce the *Grin1* knockout in most experiments. Due to a shortage of the correct vector, a subset (6 of 14) of the ΔGrin1_{adult} experiments were performed without the mCherry fluorophore. For pharmacological NMDA receptor inhibition experiments (*Figure 2—figure supplement*

*Figure 2—figure supplement 1C and D*), adult C57BL/6 mice were injected with 0.1 mg/kg MK-801 intraperitoneally and neuronal activity was recorded before and after injection.

## Optogenetic activation of paAIP2 and laser attenuation measurements

We used a photoactivatable autocamtide inhibitory peptide 2 (paAIP2) (*Murakoshi et al., 2017*) to inhibit CaMKII for the entire duration of the visuomotor exposure in the virtual reality environment. We directed a blue laser (OBIS 473 nm LX 75 mW, Coherent) onto V1 in both hemispheres using a galvo-galvo system and (GVSM002-EC/M, Thorlabs). Beam diameter on the cortical surface was 3 mm (FWHM). Light was triggered at 0.2 Hz with a duty cycle of 20% (1 s on, 4 s off). During illumination periods, we alternated between the two hemispheres at 50 Hz. Peak laser power was 20 mW, which resulted in a time-averaged power density at the cortical surface of 0.28 mW/mm$^2$. To measure the laser attenuation through tissue (*Figure 6—figure supplement 1A*), we prepared slices of fresh brain tissue of 100 μm, 200 μm, 300 μm, 400 μm, and 500 μm thickness. We then illuminated slices with the blue laser used for optogenetic inhibition of CaMKII set to 20 mW power and measured the fraction of power transmitted through each slice using a power meter (PM100D, Thorlabs).

## Single-nuclei RNA sequencing

Two *Grin1$^{tm2Stl}$* mice (P21) were anesthetized, as described above, and injected with either AAV2/1-EF1α-Cre-T2A-mCherry or AAV2/1-EF1α-mCherry-IRES-Flpo at P21. Two injections, ~250 nl each, were made in V1 bilaterally. Mice were sacrificed for single-nuclei RNA sequencing between P41 and P45. Prior to nuclei isolation, mice were again anesthetized and perfused with carbogenated, ice-cold choline ACSF (92 mM choline-Cl, 30 mM NaHCO$_3$, 5 mM Na-ascorbate, 10 mM MgSO$_4$, 3 mM Na-pyruvate, 2.5 mM KCl, 1.2 mM NaH$_2$PO$_4$, and 0.5 mM CaCl$_2$, pH 7.35, 310 mOsm). After perfusion, mice were decapitated, and the brain was removed and placed into a Petri dish containing ice-cold choline solution. The neocortex was separated from the rest of the brain, and using an epifluorescent microscope, an area corresponding to V1 with visible mCherry expression was dissected and placed into a 1.5 ml RNase-free Eppendorf tube. The tube was then flash frozen in liquid nitrogen and stored at –80°C prior to nuclei isolation.

Nuclei isolation proceeded as previously described (*Corces et al., 2017*), with the modifications described in the following. In brief, dissected brain tissue was first thawed on ice for 5 min and 2 ml of homogenization buffer was added (250 mM sucrose, 25 mM KCl, 5 mM MgCl$_2$, 20 mm Tricine-KOH, 1 mM DTT, 0.5 mM spermidine, 0.15 mM spermine, 1 protease inhibitor tablet [cOmplete Protease Inhibitor Cocktail, Roche] per 50 ml, 0.07% RNAse inhibitor [Promega], 0.4% bovine serum albumin [BSA]). The solution was then dounce homogenized 10 times with a loose pestle, 50 μl of 6% IGEPAL-630 was added to each sample, followed by homogenization with a tight pestle 5 times. The homogenized solution was passed through a 40 μm filter (Flowmi cell strainer) and combined with an equal volume of 50% iodixanol in diluent buffer (150 mM KCl, 30 mM MgCl$_2$, 120 mM Tricine-KOH pH 7.8). Samples were each transferred to a 13.2 ml Ultra-clear centrifuge tube (Beckman Coulter) and underlayered with 30% and then 40% iodixanol. Samples were centrifuged at 10,000 × *g* for 18 min at 4°C in a prechilled swinging bucket rotor. The interface between the 30 and 40% iodixanol solutions was collected. Then DAPI (1:1000) and DRAQ5 (Thermo Fisher) (1:200) were added to the sample fractions, and each sample was passed through a 30 μm filter (CellTrics).

Isolated fractions were immediately sorted with an MA900 cell sorter (Sony) using a 100 μm chip. To exclude debris, particles were selected by shape (back and forward scatter) to enrich the nuclei population. DRAQ5 and DAPI stains were also used during sorting to ensure purity and exclude doublets. Typically, between 70 and 90% of all events were nuclei. At the early stages of experiment development, samples were stained with DAPI and DRAQ5 and examined using fluorescence microscopy to confirm that sorted samples were enriched for nuclei. Samples were sorted directly into a PBS and BSA solution, and the final concentration of the BSA was 0.04%. In all instances, 14,000 nuclei were sorted. Samples were processed with the Chromium Single Cell Gene Expression kit. Gel bead-in-emulsion generation, reverse transcription, barcoding, cDNA amplification and purification were all in accordance with the manufacturer's recommendations. Finalized libraries were sequenced on a NextSeq machine using the High-out 75-cycle paired-end protocol, to a read depth of approximately 15,000–20,000 reads per nucleus.

## Single-RNA nuclei-sequencing analysis

Initial processing was performed with the Cell Ranger software package (version 6.1.2). Mapping was performed against a custom genome and included intronic reads. The custom genome was constructed from the Genome Reference Consortium Mouse Build 39 with a version 104 GTF file. For mapping viral expression, an mCherry sequence (Addgene 237633) was amended as a separate chromosome. The GTF file was edited to include features corresponding to viral expression, as well as a portion of chromosome 2 (25,179,192–25,190,563 bp), which corresponds to the part of the *Grin1* gene that is flanked by loxP sites in the *Grin1^tm2Stl* mice. Features on the minus strand of this region were removed from the analysis.

Raw feature barcode matrices were imported into R using the Scater package (*McCarthy et al., 2017*). Cells were initially identified with DropletUtils (*Lun et al., 2019*) with a barcode rank threshold of 500 and an FDR value of 0.001. Subsequent analyses were performed with Seurat version 3.0 (*Stuart et al., 2019*) in conjunction with custom-written R scripts. Additionally, to properly establish cell identity in the nuclei dataset, the higher depth dataset from the Allen Institute for Brain Science (*Tasic et al., 2018*) was used as a basis of comparison for marker expression and cluster identities. All samples, including data from the Allen dataset, were projected into the same low-dimensional space using the LIGER package (*Welch et al., 2019*) with a Seurat wrapper to calculate iNMF vectors. 10,000 features were selected using the variance-stabilizing transformation method. Data integration used a K value of 30 and a lambda of 1. Additionally, the data were also clustered using a graph-based approach (*Macosko et al., 2015*) with a resolution of 0.3. Next, a weighted nearest-neighbor analysis was performed using iNMF vectors to determine the identity of the closest cell group in the Allen dataset for all nuclei in our dataset. For this analysis, we calculated a distance weighted mean of the 10 nearest neighbors. A cutoff value was used to exclude cells that did not clearly map to any given cell type defined by the Allen dataset. This was necessary as some of the samples also contained small fractions of cells from subcortical tissue. We noticed that the VIP and SST groups were misassigned by the LIGER algorithm and corrected this by assigning these clusters to their correct identity based on marker expression.

## Calcium imaging data analysis

Calcium imaging data were processed as described previously (*Keller et al., 2012*). In brief, raw images were full-frame registered to correct for lateral brain motion. Neurons were selected manually based on mean and maximum fluorescence images. Average fluorescence per neuron over time was corrected for slow fluorescence drift using an 8th percentile filter and a 100 s window (*Dombeck et al., 2007*) and divided by the median value over the entire trace to calculate ΔF/F.

Data analysis was performed with custom analysis scripts written in MATLAB 2020b (MathWorks). For all population onset responses, data were first averaged over onsets for each neuron and then averaged over neurons. Unless stated otherwise, shading and error bars indicate the standard error of the mean (SEM) across neurons. We did not test for normality of distributions. For analysis of onset responses (*Figure 2A-C*, *Figure 3A-C*, *Figure 4A and B*, *Figure 6C-H*, *Figure 6—figure supplement 2C-E*, *Figure 6—figure supplement 3A-F*), recording sites with less than three running or mismatch onsets in a particular session were excluded from analysis (e.g., if a mouse ran without stopping for the entire duration of a closed-loop session, there were no running onsets to analyze). We also excluded two sessions in which the mouse did not run without prompting by the experimenter. In total, we excluded 10 of 384 sessions. In 31 of the remaining sessions, we did not record grating responses. To calculate stimulus-induced changes in ΔF/F, we used a baseline subtraction window of –300 ms to 0 ms, and a response window of +100 ms to +1500 ms relative to stimulus onset. To determine running onsets, we used a threshold of $10^{-2}$ cm/s. For analysis of average activity levels in closed-loop sessions (*Figure 2D*, *Figure 2—figure supplement 1C*, *Figure 3D*), we calculated the average neuronal activity (ΔF/F [%]) over time and over neurons. To calculate average pairwise correlation between neurons in the closed-loop condition (*Figure 2F*, *Figure 2—figure supplement 1D*, *Figure 3F*, *Figure 6—figure supplement 1B*), we calculated the mean correlation of each neuron's activity with all other neurons' activity. To calculate the first principal component (*Figure 2E*, *Figure 2—figure supplement 1B*, *Figure 3E*), we calculated the eigenvectors of the covariance matrix of the mean-subtracted visual flow and running correlations with neuronal activity. The principal component angle was defined as the angle between the first principal component and the y axis.

## Acknowledgements

We thank Tara Keck for valuable comments on the manuscript, all members of the Keller lab for discussion and support, and Tingjia Lu and Daniela Gerosa for vector production. We thank Ryohei Yasuda for valuable advice and help with paAIP2. We thank Sirisha Aluri, Sebastien Smallwood, and Hubertus Kohler for help with the single-nuclei RNA-sequencing experiments. This project has received funding from the Swiss National Science Foundation, the Novartis Research Foundation, and the European Research Council (ERC) under the European Union's Horizon 2020 research and innovation programme (grant agreement no. 865617).

## Additional information

### Funding

| Funder | Grant reference number | Author |
|---|---|---|
| Schweizerischer Nationalfonds zur Förderung der Wissenschaftlichen Forschung | | Georg B Keller |
| H2020 European Research Council | 865617 | Georg B Keller |

The funders had no role in study design, data collection and interpretation, or the decision to submit the work for publication.

### Author contributions

Felix C Widmer, Conceptualization, Data curation, Formal analysis, Investigation, Methodology, Software, Validation, Visualization, Writing – original draft, Writing – review and editing; Sean M O'Toole, Conceptualization, Data curation, Formal analysis, Writing – review and editing; Georg B Keller, Funding acquisition, Supervision, Writing – original draft, Writing – review and editing

### Author ORCIDs

Felix C Widmer http://orcid.org/0000-0002-9045-8909
Georg B Keller http://orcid.org/0000-0002-1401-0117

### Ethics

All animal procedures were approved by and carried out in accordance with guidelines of the Veterinary Department of the Canton Basel-Stadt, Switzerland under license number 2573.

### Decision letter and Author response

Decision letter https://doi.org/10.7554/eLife.71476.sa1
Author response https://doi.org/10.7554/eLife.71476.sa2

## Additional files

### Supplementary files

- Supplementary file 1. Statistical information. (A) Table of all statistical information for all figure panels that display the results of statistical testing. (B) List of the number of mice of each genotype for each experiment.
- Transparent reporting form

### Data availability

Software for controlling the two-photon microscope and preprocessing of the calcium imaging data is available on https://sourceforge.net/projects/iris-scanning/. Raw data and code to generate all figures of this manuscript are available on https://data.fmi.ch/PublicationSupplementRepo/.

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
