## [Editor Report]

This study demonstrates that the development of visuomotor mismatch signals in V1 as well as behavioral reactions to mismatches depend on NMDA receptors in V1 during early visual experience, suggesting a critical role of NMDA receptor-dependent plasticity within V1 in forming internal models that transform self-movement to visual experiences. The result will be of great interest to the neuroscience community.

---

## [Decision Letter]

**Decision letter after peer review:**

Thank you for submitting your article "Developmental plasticity in visual cortex is necessary for normal visuomotor integration and visuomotor skill learning" for consideration by *eLife*. Your article has been reviewed by 2 peer reviewers, and the evaluation has been overseen by a Reviewing Editor and Andrew King as the Senior Editor. The reviewers have opted to remain anonymous.

Essential revisions:

This study examined the mechanism underlying the development of prediction-error related responses of neurons in the primary visual cortex evoked by mismatch between self-generated locomotor movement and visual feedback. The reviewers found that this study contains important results, in particular, a local mechanism involving NMDA receptors and CaMKII. The manuscript is written clearly and discusses various limitations of the study carefully. However, the reviewers have made various points that need to be clarified or discussed in more detail, and made several suggestions that will likely improve this work. We do not expect that addressing these issues will necessarily require additional experiments, although some points may benefit from them.

In the following, we list a few essential points that the reviewers have identified. The detailed comments associated with these points are further described in the individual reviewers' comments below.

1) Both reviewers raised the issue of the uncertainty of the site of action of NMDA receptors and CaMKII. The reviewers suggest some clarifications of which cell types or layers express Cre-recombinase used for conditional knockouts and some verification of gene knockouts. Are there any relationships between cellular responses and these factors? We recognize that these analyses can be difficult, but if the authors can include some data or discussion that addresses these points, it would be great to present them.

2) The mismatch signals vary across "control" groups in different experiments. It is worrisome that this might be leading to inaccurate characterizations of the effects of particular manipulations. See Reviewer 1's comment for more details.

3) From the presented data, it is difficult to know whether the altered neuronal responses and behaviors are due to a deficit in plasticity. While this is not an unreasonable argument, given the difficulty of excluding other mechanisms (e.g. bottom-up activation), it would be good to tone down the claim of plasticity (particularly as it is in the title).

*Reviewer #1 (Recommendations for the authors):*

This is a really nice set of experiments that help us understand where in the brain plasticity is happening as mice learn the sensory consequences of their actions. I think the question is very important and that the particular perturbations that were performed and the analyses are all appropriate. I also commend the authors on their recognition of the limits of their study in the Discussion – the limits are all reasonable. I don't have any new experiments that I would recommend, but see a suggested analysis (next paragraph) and a few other smaller points.

Presumably not all V1 neurons expressed Cre-recombinase (which was necessary to interfere with NMDAR function). What did the distribution of mismatch signals look like in V1 of Grin1 mice? Do some neurons show perfectly normal mismatch signals? Taking that just a bit further, is there enough statistical power to analyze whether there is any correspondence between the magnitude of mismatch signals and the active/inactive status of NMDARs at the single-cell level? I recognize that this might be a difficult analysis to do with low N's of trials. Also, it's quite possible that what we're observing here is network-level learning, and that you may not see cell-to-cell changes in mismatch signals that correspond with NMDAR suppression (presumably a binary value). However, given the findings of excitatory-cell-specific CaMKII inactivation, it's plausible that the function of NMDARs within single L2/3 cells might allow for the continued single-cell computation of mismatch signals to persist, even in the absence of functional NMDARs in neighboring cells. Therefore, if there was a correspondence between NMDAR function and mismatch at the single cell level, this would be quite strong evidence pointing toward the importance of functional NMDARs in single cells. The authors *do not need* to do this analysis for me to recommend acceptance of this paper. I simply bring it up because I think it is a potential analysis that could help solidify the importance of NMDARs.

The mismatch signals across different "control" groups in this paper are quite different, and I worry that perhaps this is leading to inaccurate characterizations of the effects of particular manipulations. In particular, the mismatch signal in the control group from the paAIP2-PV experiments (Figure S3C) is almost non-existent. These data are used to show that blocking CaMKII in PV cells leads to an increased mismatch responses and is therefore different from blocking CaMKII in SST cells (which leads to a decreased mismatch response, Figure 5F). But if these mice were compared against a different control group, there could be either no effect or the exact opposite effect. I would find it helpful if the authors spoke to the variability in results from distinct control groups in the results or the discussion. Alternatively, the authors could show that their findings hold up to some other analysis in which a more consistent control condition is used (if possible).

Line 109 describes correlations between neural activity and optic flow and locomotion. These findings make sense to me if they were analyzed in the open-loop variant, but I did not see in either the Methods or the Results whether we switched to a new experiment at this point (the previous data were presumably about closed loop).

Line 205 begins to describe the optical strategy for suppression of NMDAR function. When I got to line 314 I learned that the blue illumination was delivered during all motor-sensory experience, but line 205 led me to believe that brief blue light led to a permanent shift. This was particularly confusing when I got to line 257, which told me that we could study the same mice without CaMKII inhibition. The authors should do a bit more methodological hand holding in the results and in the Methods.

Line 138, did the adult Grin1 mice still have their NMDARs silenced when the VR acclimation/training started? It's unclear from the paper. (I think this is the case but would be great to have it spelled out).

In Figure 4E, I can't figure out what each dot represents. At first I thought each dot was a mouse, but it seems like too many dots for the number of mice included in the study groups. Once I know what the dots represent, I will be interested in the overlap across populations. For example, many of the Grin1-Juv dots are above the Grin1-adult dots and are on par with the Control-Juv dots. Is there an explanation for this? Was the NMDAR perturbation more quantifiably efficacious for some mice than for others?

*Reviewer #2 (Recommendations for the authors):*

Major issues

1) Presented data do not convincingly support the direct role of NMDAR or CamK2 dependent plasticity for visuomotor integration associated with first visual experience. As grating onset response was reduced by juvenile V1 NMDAR knock-out or CaMK2 manipulations in addition to mismatch onset response, it is totally possible that diminished mismatch response is secondary to diminished visual response due to plasticity deficits of bottom-up visual input connectivity only without requiring plasticity in bottom-up/top-down integration. This possibility needs to be discussed, and title of the paper and overall interpretation of the data should be revised accordingly.

2) Manipulations of Grin and CaMK2 led to very different V1 responses for grating onset response (Fig1F vs Fig5DG). One possibility not discussed was that manipulation in single cell type (Figure 5) vs multiple cell types (Figure 1 uses non-cell type selective promotor for cre expression) might lead to different outcomes. Ideally, it would be informative to manipulate Grin and CaMK2 in same cell types to make better comparisons of the two mechanisms. At least, it would be informative to check if Grin KO is happening in both pyramidal and SST cells in Figure 1.

3) At the level of activity correlation visual flow, results shown in Fig6 are consistent with the unique role of SST in contributing to the mismatch response deficits. To increase the coherence of the data among different experiments, I suggest showing data in Fig6A as in Fig1I and Fig2E. Similarly, Fig1I and Fig2E can include bar graphs similar to Fig6A.

Other points

4) Quantification of Grin knock-down is missing for both juvenile (Figure 1) and adult (Figure 2) knock out studies. While one representative image is shown in Figure 1, knock down level is not quantified. Quantification in the adult mice is also important as Grin1 is likely less expressed compared to young mice. It is also not described at what age AAV was injected and at what age animals were sacrificed for in situ hybridization. It would be important to match the age to the experimental conditions to ensure timely knock-down.

5) Comparison between dark rearing+Juvenile Grin knock out condition (Figure 1) vs normal rearing+adult Grin KO (Figure 2) is not straight forward without another control condition (normal rearing+juvenile Grin1 knock out). This limitation should be discussed.

6) Dark rearing of mice from birth is known to delay the timecouse of V1 critical period by keeping V1 circuits immature. It is unclear how such changes induced by dark rearing confound the findings reported in this study. Without dark rearing, it is likely that the physiological experience-dependent sensory-motor integration happens earlier during development. I suggest to discuss these points in Discussion section.

7) Animal number information should be provided for each experiments instead of grouping together in Table S2 to ensure that data is not biased from limited number of mice.

---

## [Author Response]

Essential revisions:This study examined the mechanism underlying the development of prediction-error related responses of neurons in the primary visual cortex evoked by mismatch between self-generated locomotor movement and visual feedback. The reviewers found that this study contains important results, in particular, a local mechanism involving NMDA receptors and CaMKII. The manuscript is written clearly and discusses various limitations of the study carefully. However, the reviewers have made various points that need to be clarified or discussed in more detail, and made several suggestions that will likely improve this work. We do not expect that addressing these issues will necessarily require additional experiments, although some points may benefit from them.In the following, we list a few essential points that the reviewers have identified. The detailed comments associated with these points are further described in the individual reviewers' comments below.1) Both reviewers raised the issue of the uncertainty of the site of action of NMDA receptors and CaMKII. The reviewers suggest some clarifications of which cell types or layers express Cre-recombinase used for conditional knockouts and some verification of gene knockouts. Are there any relationships between cellular responses and these factors? We recognize that these analyses can be difficult, but if the authors can include some data or discussion that addresses these points, it would be great to present them.

This is an excellent question and we had previously attempted to address this by looking for a relationship between mCherry expression and functional effects in the knockout data (the vector used to induce the knockout was an AAV2/1-EF1α-Cre-T2A-mCherry). However, we failed to detect any systematic relationship. We speculated that this might be the consequence of the fact that the knockout is non-linearly related to Cre expression (in principle, one Cre molecule suffices for a complete knockout), while the mCherry measurement is linear in expression level. Thus, it is possible that even in cells in which we did not detect any mCherry, there was a complete knockout. On top of this, there is a correlation between mCherry and GCaMP responses that is driven by differences in imaging quality that interferes with this analysis (see also response to Review 1 point 1, below).

To determine the strength of the knockout, we first quantified the in-situ hybridization data. We selected a random set of 500 µm by 500 µm regions either at injection sites or away from injection sites and counted the number of cells that stained positive for *Grin1* mRNA (using an in-situ hybridization probe that targets a region of mRNA that is within the region of the *Grin1* gene that is knocked out). Even with a very low threshold for what we counted as “*Grin1*-positive” (more than 2 puncta – see Figure 1E (ii) for what we define as a punctum), we found that less than 1% of the cells at an injection site were still *Grin1* positive.

To confirm this result, and to test whether the knockout spared certain neuron types, we performed a single nuclei sequencing experiment (new Figure 1G). Mice received an injection of either Cre (knockout) or Flp (control) AAV vectors at P21, before being sacrificed between P41 and P45 for single nuclei sequencing. We were able to identify all major cortical neuron types in both knockout and control data (new Figure 1—figure supplement 1) and found that the knockout reduced *Grin1* expression in all neuron types. Please note, we found some remaining *Grin1* reads also in the knockout samples, but we suspect this is mainly the consequence of the dissected tissue also containing cells that were outside the region of knockout.

2) The mismatch signals vary across "control" groups in different experiments. It is worrisome that this might be leading to inaccurate characterizations of the effects of particular manipulations. See Reviewer 1's comment for more details.

There are likely two things that contributed to this variability across control groups. First, population average mismatch responses by imaging site are quite variable (see Author response image 1). We were aware of this problem and for this reason chose to design these experiments with a within-animal control. Second, something we did not foresee, we suspect that there are compensatory mechanisms that altered responses in the control hemispheres. These effects were primarily apparent in the paAIP2 data.

**Author response image 1. sa2fig1:** Mismatch responses per mouse. Black lines are the average mismatch responses over mice, gray lines are the mismatch responses of individual mice (A) Data of coupled trained (CT) mice of the (Attinger et al., 2017) data. (B) Data of the ΔGrin1_juv_ mice. (C) Data of the ΔGrin1_adult_ mice. Representation as in Figure 2A.

In Author response image 2, we compared all effects to different control groups. We used the control hemisphere data of the *ΔGrin1_juv_* mice and the *ΔGrin1_adult_* mice and refer to these as “juvenile control” and “adult control” in the following passage. The overall effects in the *ΔGrin1* data were much smaller than in the paAIP2 data, and consequently any putative interhemispheric compensation is also likely smaller. A few findings may stand out. First, mismatch responses were larger in the adult mice than in juvenile mice, but this is perhaps not surprising as mismatch responses increased with experience (Attinger et al., 2017). Second, in the group of juvenile mice that received CaMKII inhibition in excitatory neurons, all responses appeared to be much faster in rise time (in both inhibited and control hemispheres). Why this is, we do not know, however, the key finding here is that mismatch responses in the inhibited hemisphere were smaller than those in the control hemisphere. While only just, this was also true if we compared peak amplitude against juvenile control. However, given that the control hemisphere responses were clearly also influenced by the inhibition of paAIP2 in the contralateral hemisphere, we suspect the better comparison is the within mouse control. Third, in mice that received CaMKII inhibition in PV interneurons, the control hemisphere mismatch responses were substantially reduced compared to those in the juvenile control. But again, the paAIP2 inhibited mismatch response was larger than both that in the control hemisphere as well as that in the juvenile control. Thus, while there are many aspects of these effects we do not fully understand yet, we don’t think we are inaccurately characterizing the mismatch response changes by using the within animal controls. We have expanded on the discussion of these problems in the manuscript.

**Author response image 2. sa2fig2:** All plots as in the manuscript but overlaying additionally the data of the ΔGrin1_juv_ and the ΔGrin1_adult_ data. (A) Data of ΔGrin1_juv_ mice (B) Data of ΔGrin1_adult_ mice, (C-E) Data for all three paAIP2 datasets. Representation as in Figures 2A-2C.

3) From the presented data, it is difficult to know whether the altered neuronal responses and behaviors are due to a deficit in plasticity. While this is not an unreasonable argument, given the difficulty of excluding other mechanisms (e.g. bottom-up activation), it would be good to tone down the claim of plasticity (particularly as it is in the title).

We have changed the title and abstract of the manuscript as suggested and now discuss alternative mechanisms more thoroughly.

Reviewer #1 (Recommendations for the authors):This is a really nice set of experiments that help us understand where in the brain plasticity is happening as mice learn the sensory consequences of their actions. I think the question is very important and that the particular perturbations that were performed and the analyses are all appropriate. I also commend the authors on their recognition of the limits of their study in the Discussion – the limits are all reasonable. I don't have any new experiments that I would recommend, but see a suggested analysis (next paragraph) and a few other smaller points.

We thank the reviewer for the help in improving the manuscript. Please note, to ease reference, we have added numbering to the following points.

Presumably not all V1 neurons expressed Cre-recombinase (which was necessary to interfere with NMDAR function). What did the distribution of mismatch signals look like in V1 of Grin1 mice? Do some neurons show perfectly normal mismatch signals? Taking that just a bit further, is there enough statistical power to analyze whether there is any correspondence between the magnitude of mismatch signals and the active/inactive status of NMDARs at the single-cell level? I recognize that this might be a difficult analysis to do with low N's of trials. Also, it's quite possible that what we're observing here is network-level learning, and that you may not see cell-to-cell changes in mismatch signals that correspond with NMDAR suppression (presumably a binary value). However, given the findings of excitatory-cell-specific CaMKII inactivation, it's plausible that the function of NMDARs within single L2/3 cells might allow for the continued single-cell computation of mismatch signals to persist, even in the absence of functional NMDARs in neighboring cells. Therefore, if there was a correspondence between NMDAR function and mismatch at the single cell level, this would be quite strong evidence pointing toward the importance of functional NMDARs in single cells. The authors do not need to do this analysis for me to recommend acceptance of this paper. I simply bring it up because I think it is a potential analysis that could help solidify the importance of NMDARs.

This is an interesting point and we had previously attempted to perform an analysis along the lines of what the reviewer is suggesting. We split the neurons based on the expression levels of mCherry (the Cre vector we used co-expressed mCherry) to test whether neurons with high levels of mCherry exhibited larger effects than those with low or no detectable expression of mCherry. The argument being – as the reviewer suggests – that neurons that expressed mCherry have a knockout of NMDA receptors while those that did not, had no knockout. The issue we encountered was that there was a positive correlation between mCherry expression and GCaMP signal strength that dominated the effect (i.e. the neurons with the lowest mCherry signals often also had low GCaMP signals – likely as a consequence of differences in local image quality – not a true co-variation of expression levels). We could not find a way to disentangle this effect with a potential cell specific effect of the knockout. In addition, given that the Cre effect is non-linear (one Cre protein suffices for the knockout, while the mCherry signal is linear in number of proteins), we thought it conceivable that the knockout may be complete in that it affected all neurons at the injection site of the Cre vector.

What we then proceeded to do, was estimate the fraction of neurons in an injection site that were affected by the knockout. We first quantified this in our in-situ hybridization data (new Figure 1F) and found that the number of cells positive for the *Grin1* exon targeted by the knockout in the injection site was below 1%. To confirm that all neuron types were affected equally by this manipulation, we then performed single nuclei sequencing on knockout tissue and found that *Grin1* mRNA levels were reduced homogeneously across all major cortical neuron types (new Figure 1G).

While we do think that the effect is likely a mixture of network and neuron specific effects, we refrained from speculating one way or the other given that the knockout appeared to be quite complete.

The mismatch signals across different "control" groups in this paper are quite different, and I worry that perhaps this is leading to inaccurate characterizations of the effects of particular manipulations. In particular, the mismatch signal in the control group from the paAIP2-PV experiments (Figure S3C) is almost non-existent. These data are used to show that blocking CaMKII in PV cells leads to an increased mismatch responses and is therefore different from blocking CaMKII in SST cells (which leads to a decreased mismatch response, Figure 5F). But if these mice were compared against a different control group, there could be either no effect or the exact opposite effect. I would find it helpful if the authors spoke to the variability in results from distinct control groups in the results or the discussion. Alternatively, the authors could show that their findings hold up to some other analysis in which a more consistent control condition is used (if possible).

We have expanded on the discussion of variability in the control groups as suggested. Please also see response to essential review comment 2 above related to this point.

Line 109 describes correlations between neural activity and optic flow and locomotion. These findings make sense to me if they were analyzed in the open-loop variant, but I did not see in either the Methods or the Results whether we switched to a new experiment at this point (the previous data were presumably about closed loop).

Please excuse the inaccuracy – the reviewer is absolutely correct. These analyses were done on the open-loop data. We have rephrased the corresponding passage to correct this.

Line 205 begins to describe the optical strategy for suppression of NMDAR function. When I got to line 314 I learned that the blue illumination was delivered during all motor-sensory experience, but line 205 led me to believe that brief blue light led to a permanent shift. This was particularly confusing when I got to line 257, which told me that we could study the same mice without CaMKII inhibition. The authors should do a bit more methodological hand holding in the results and in the Methods.

We have rephrased and expanded as suggested.

Line 138, did the adult Grin1 mice still have their NMDARs silenced when the VR acclimation/training started? It's unclear from the paper. (I think this is the case but would be great to have it spelled out).

This is correct – the adult group of *Grin1* KO mice had received a *Grin1* knockout prior to the start of training on the VR setup (but have normal *Grin1* levels up to that point). We have changed the passage to make this clearer as suggested.

In Figure 4E, I can't figure out what each dot represents. At first I thought each dot was a mouse, but it seems like too many dots for the number of mice included in the study groups. Once I know what the dots represent, I will be interested in the overlap across populations. For example, many of the Grin1-Juv dots are above the Grin1-adult dots and are on par with the Control-Juv dots. Is there an explanation for this? Was the NMDAR perturbation more quantifiably efficacious for some mice than for others?

This is a very good catch – each dot should indeed have been one mouse. As a result of a bug in our analysis code, most mice were included twice. “Early” and “late” timepoints are days 1 and 2, and days 6 and 7, respectively. For most of the mice the two time points were not combined but plotted separately. Please excuse the mistake.

Regarding the variability – this is likely driven predominantly by mouse-to-mouse variability in this metric. The variability in Control_juv_ mice was similar to that in *ΔGrin1_juv_* mice. Thus, even without any variability in the effect size of the knockout, we would expect to have roughly this level of variability in *ΔGrin1_juv_* mice. To quantify the relationship between knockout efficacy and effect size would require substantially larger cohorts (or an assay with less mouse-to-mouse variability).

Reviewer #2 (Recommendations for the authors):

Major issues

1) Presented data do not convincingly support the direct role of NMDAR or CamK2 dependent plasticity for visuomotor integration associated with first visual experience. As grating onset response was reduced by juvenile V1 NMDAR knock-out or CaMK2 manipulations in addition to mismatch onset response, it is totally possible that diminished mismatch response is secondary to diminished visual response due to plasticity deficits of bottom-up visual input connectivity only without requiring plasticity in bottom-up/top-down integration. This possibility needs to be discussed, and title of the paper and overall interpretation of the data should be revised accordingly.

This is absolutely correct, however, there are a few things we would like to note. First, to avoid a potential misunderstanding, we are working under the assumption that all layer 2/3 neurons establish an opposing balance between top-down and bottom-up input (Jordan and Keller, 2020). Second, we used the concept of plasticity in the integration of bottom-up and top-down input to mean plasticity in either top-down input or bottom-up input, or both. The system needs to match opposing bottom-up and top-down inputs in L2/3 neurons to compute prediction error responses, but this can happen either by matching bottom-up input to a fixed top-down input, or vice versa, or by changing both. Thus, a deficit in bottom-up input connectivity onto L2/3 neuron would be consistent with our interpretation. An alternative, possibly along the lines of what the reviewer was suggesting, is that the NMDA knockout mediates its effects by virtue of reducing feed-forward input onto layer 4 neurons. We think this is unlikely as a knockout of NMDA receptors in L4 neurons does not alter visually evoked potentials in visual cortex, nor does it impair visual acuity of the mice, regardless of whether the knockout is congenital or postadolescent (Fong et al., 2020; Sawtell et al., 2003). We have rephrased and revised the manuscript to make this clearer.

2) Manipulations of Grin and CaMK2 led to very different V1 responses for grating onset response (Fig1F vs Fig5DG). One possibility not discussed was that manipulation in single cell type (Figure 5) vs multiple cell types (Figure 1 uses non-cell type selective promotor for cre expression) might lead to different outcomes. Ideally, it would be informative to manipulate Grin and CaMK2 in same cell types to make better comparisons of the two mechanisms. At least, it would be informative to check if Grin KO is happening in both pyramidal and SST cells in Figure 1.

To address the question of which cell types are targeted by the *Grin1* knockout, we have added the experiments and analyses shown in Figures 1F and 1G. In brief, the knockout seems to indeed affect all cell types similarly. This is likely due to the fact that the Cre effect is highly non-linear (i.e. one Cre protein suffices for a complete knockout), and that Ef1α promoter drives expression also in interneurons, albeit at low levels. The 1.3kb version of the CaMKII promoter we used to express the paAIP2 in excitatory neurons also drives expression in low levels in inhibitory neurons, similar to Ef1α(Scheyltjens et al., 2015). However, the effect of CaMKII inhibition is probably linear in the number of paAIP2 molecules, and hence indeed biased towards excitatory neurons. We expanded on this point in the discussion.

3) At the level of activity correlation visual flow, results shown in Fig6 are consistent with the unique role of SST in contributing to the mismatch response deficits. To increase the coherence of the data among different experiments, I suggest showing data in Fig6A as in Fig1I and Fig2E. Similarly, Fig1I and Fig2E can include bar graphs similar to Fig6A.

We chose not to present the data this way as we feel it distracts from the main points. The average visual flow correlation differences (shown in previous Figure 6A, current Figure 7A) are barely visible in the correlation scatter plots shown in Figures 2E and 3E (previous Figures 1I and 2E). Conversely, the visual flow correlations of the NMDA knockout data only become relevant in the comparison to the correlations of the paAIP2 SST data, and this is all in Figure 7A (previous Figure 6A). We have added the scatter plots to Author response image 3 but have not added them to the manuscript as we think it would disrupt the flow and add only little to the manuscript.

**Author response image 3. sa2fig3:** Scatter plots of the correlation between neuronal activity and visual flow, and the correlation between neuronal activity and running speed in open-loop sessions for (A) adult control mice, (B) the ΔGrin1_juv_ and ΔGrin1_adult_ dataset, and (C) all three paAIP2 datasets. Corresponding to Figures 2E and 3E.

Other points4) Quantification of Grin knock-down is missing for both juvenile (Figure 1) and adult (Figure 2) knock out studies. While one representative image is shown in Figure 1, knock down level is not quantified. Quantification in the adult mice is also important as Grin1 is likely less expressed compared to young mice. It is also not described at what age AAV was injected and at what age animals were sacrificed for in situ hybridization. It would be important to match the age to the experimental conditions to ensure timely knock-down.

We have added the information as requested and have added a more detailed quantification of the knockout using both in situ hybridization data and single nuclei sequencing data. See response to essential review point 1 in the summary above. With respect to the age matching of the knockout quantification, in the in situ analysis we have data from 2 mice, one was age matched to the juvenile experiments and one was age matched to the adult. The analysis shown in Figure 1F, split by the juvenile and adult, is shown in Author response image 4. Given that we did not find any evidence of a difference in these data and could not find any published evidence to suggest that the knockout could be differentially effective in these two age groups, we pooled all data for the analysis shown in Figure 1F.

**Author response image 4. sa2fig4:** Analysis of Figure 1F split by age, matched to the two conditions used for the functional experiments, juvenile and adult. Note, the data are based on multiple slices and quantification regions but are from one mouse each.

5) Comparison between dark rearing+Juvenile Grin knock out condition (Figure 1) vs normal rearing+adult Grin KO (Figure 2) is not straight forward without another control condition (normal rearing+juvenile Grin1 knock out). This limitation should be discussed.

We have added this point to the discussion as suggested.

6) Dark rearing of mice from birth is known to delay the timecouse of V1 critical period by keeping V1 circuits immature. It is unclear how such changes induced by dark rearing confound the findings reported in this study. Without dark rearing, it is likely that the physiological experience-dependent sensory-motor integration happens earlier during development. I suggest to discuss these points in Discussion section.

This is correct, the dark rearing in our paradigm likely influences the timing of critical windows for plasticity. However, please note that the paradigm used in the current study was the same as that used in a previous study (Attinger et al., 2017). There we show that this paradigm (dark rearing plus virtual reality training experience) resulted in increased prediction error responses in V1 (compared to normally reared mice). Thus, while there likely are changes in critical window timing, this should not change our conclusions we draw based on the data. We have added this point to the discussion as suggested.

7) Animal number information should be provided for each experiments instead of grouping together in Table S2 to ensure that data is not biased from limited number of mice.

We have added information on the number of mice for all experiments to Supplementary Table 1 (in parenthesis in the columns N_1_ and N_2_).

**References**

Attinger, A., Wang, B., and Keller, G.B. (2017). Visuomotor Coupling Shapes the Functional Development of Mouse Visual Cortex. Cell *169*, 1291-1302.e14.

Fong, M.F., Finnie, P.S., Kim, T., Thomazeau, A., Kaplan, E.S., Cooke, S.F., and Bear, M.F. (2020). Distinct Laminar Requirements for NMDA Receptors in Experience-Dependent Visual Cortical Plasticity. Cerebral Cortex (New York, N.Y. : 1991) *30*, 2555–2572.

Jordan, R., and Keller, G.B. (2020). Opposing Influence of Top-down and Bottom-up Input on Excitatory Layer 2/3 Neurons in Mouse Primary Visual Cortex. Neuron *108*, 1194-1206.e5.

Sawtell, N.B., Frenkel, M.Y., Philpot, B.D., Nakazawa, K., Tonegawa, S., and Bear, M.F. (2003). NMDA receptor-dependent ocular dominance plasticity in adult visual cortex. Neuron *38*, 977–985.

Scheyltjens, I., Laramée, M.-E., Van den Haute, C., Gijsbers, R., Debyser, Z., Baekelandt, V., Vreysen, S., and Arckens, L. (2015). Evaluation of the expression pattern of rAAV2/1, 2/5, 2/7, 2/8, and 2/9 serotypes with different promoters in the mouse visual cortex. Journal of Comparative Neurology *523*, 2019–2042.